# The extrafollicular B cell response is a hallmark of childhood idiopathic nephrotic syndrome

Tho-Alfakar Al-Aubodah [1,2,3,4,5], Lamine Aoudjit[3,5], Giuseppe Pascale[5,6], Maneka A. Perinpanayagam[7], David Langlais [1,8], Martin Bitzan[6,9], Susan M. Samuel[7], Ciriaco A. Piccirillo[1,2,4] ✉ & Tomoko Takano [3,4,5] ✉

The efficacy of the B cell-targeting drug rituximab (RTX) in childhood idiopathic nephrotic syndrome (INS) suggests that B cells may be implicated in disease pathogenesis. However, B cell characterization in children with INS remains limited. Here, using single-cell RNA sequencing, we demonstrate that a B cell transcriptional program poised for effector functions represents the major immune perturbation in blood samples from children with active INS. This transcriptional profile was associated with an extrafollicular B cell response marked by the expansion of atypical B cells (atBCs), marginal zone-like B cells, and antibody-secreting cells (ASCs). Flow cytometry of blood from 13 children with active INS and 24 healthy donors confirmed the presence of an extrafollicular B cell response denoted by the expansion of proliferating RTX-sensitive extrafollicular (CXCR5⁻) CD21^low T-bet⁺ CD11c⁺ atBCs and short-lived T-bet⁺ ASCs in INS. Together, our study provides evidence for an extrafollicular origin for humoral immunity in active INS.

Idiopathic nephrotic syndrome (INS), the most common chronic glomerular disorder in children, features recurrent episodes of heavy proteinuria caused by injury to the principal filtering cell of the glomerulus, the podocyte[1,2]. The resulting podocyte lesion is often the sole histopathological manifestation of childhood INS—termed minimal change disease (MCD)—with <20% of cases presenting as more severe focal segmental glomerulosclerosis (FSGS)[1]. While an immune etiology is predicted given the efficacy of glucocorticoids (GC) and other broadly immunosuppressive drugs at reversing podocyte injury and mediating remission from proteinuria, the precise immune mechanisms involved in INS pathogenesis remain elusive. With

frequent relapses, multiple rounds of GC administration are often warranted, resulting in substantial GC-associated toxicity. Therefore, a complete understanding of disease pathogenesis is a priority for the development of safe and targeted GC-sparing therapies[3].

The recent identification of rituximab (RTX), a CD20-targeting B cell-depleting monoclonal antibody, as an effective therapeutic option to maintain long-term remission in GC-treated individuals pointed to a previously unrecognized role for B cells in the immunopathogenesis of INS[4–7]. Indeed, several immunophenotyping studies have since demonstrated that elevated levels of circulating B cells are a robust feature of active disease in affected children and adults[8–11]. The

¹Department of Microbiology & Immunology, Faculty of Medicine and Health Sciences, McGill University, Montréal, Québec, Canada. ²Infectious Diseases and Immunity in Global Health Program, Research Institute of the McGill University Health Centre, Montréal, Québec, Canada. ³Metabolic Disorders and Complications Program, Research Institute of the McGill University Health Centre, Montréal, Québec, Canada. ⁴Centre of Excellence in Translational Immunology, Research Institute of the McGill University Health Centre, Montréal, Québec, Canada. ⁵Division of Nephrology, Faculty of Medicine and Health Sciences, McGill University, Montréal, Québec, Canada. ⁶Division of Nephrology, Department of Pediatrics, Faculty of Medicine and Health Sciences, McGill University, Montréal, Québec, Canada. ⁷Section of Nephrology, Department of Pediatrics, Cumming School of Medicine, University of Calgary, Calgary, Alberta, Canada. ⁸Department of Human Genetics, Faculty of Medicine and Health Sciences, McGill University Genome Centre, Montréal, Québec, Canada. ⁹Kidney Centre of Excellence, Al Jalila Children's Hospital, and Mohammed Bin Rashid University of Medicine and Health Sciences, Dubai, UAE. ✉e-mail: ciro.piccirillo@mcgill.ca; tomoko.takano@mcgill.ca

expansion of isotype-switched classical memory B cells (cMBCs) and a converse reduction in transitional naïve B cells in the peripheral blood denotes the involvement of a classical follicular B cell response wherein B cells undergo class-switch recombination in germinal centers and generate long-lived antibody-secreting cells (ASCs)[4,11,12]. Accordingly, ASCs are elevated in adults and children with INS and circulating autoantibodies against several podocyte autoantigens have been identified in subpopulations of affected individuals, underlining a bona fide autoimmune humoral origin for INS[13–17]. Nevertheless, the exact nature of the nephrotic B cell response remains to be investigated beyond the enumeration of broad B cell subsets.

Although B cell depletion provides long-term remission from proteinuria, many patients eventually relapse through an unknown mechanism[18,19]. Seminal work demonstrated that post-RTX relapses of INS were associated with a resurgence of isotype-switched cMBCs, indicating that follicular B cell responses may be responsible for generating autoreactive ASCs[20]. However, B cell depletion with CD20-targeting biologics does not eliminate long-lived bone marrow-residing ASCs as they are devoid of surface CD20[21]. In contrast, short-lived ASCs in the periphery, while also lacking surface CD20, are effectively depleted after RTX therapy, as the antigen-experienced B cell pools from which they arise are ablated. The impact of RTX on the ASC compartment in INS has not been investigated.

Unlike long-lived ASCs, short-lived ASCs are generated through extrafollicular B cell responses with lower degrees of class-switch recombination and somatic hypermutation than classical follicular responses[22,23]. Atypical B cells (atBCs), a population of T-bet⁺ CD11c⁺ B cells, are now recognized to be an important source for short-lived ASCs[24,25]. This population arises in chronic viral and parasitic infection and is enriched with autoreactive clones in autoimmunity[26–33]. Indeed, upper respiratory tract viral infections can precipitate and exacerbate relapses of INS[34–36]. Therefore, extrafollicular reactions giving rise to RTX-sensitive short-lived ASCs may represent a major source of podocyte-targeting antibodies in INS.

In this study, we characterize the nature of the nephrotic B cell response by defining an INS-associated B cell transcriptional signature and identifying the contributing pathogenic B cell populations in disease. Using single-cell RNA-sequencing (scRNA-seq) of peripheral blood mononuclear cells (PBMC) derived from four children with active INS and age-matched healthy controls (HC), we demonstrate that the mobilization of memory B cells through an extrafollicular route represents the major immunological abnormality in peripheral blood. Subsequent flow cytometric characterization of B cells in a cohort of 13 children with active INS and 24 HCs shows that this is associated with the expansion of RTX-sensitive CD21^low CXCR5⁻ T-bet⁺ CD11c⁺ atBCs and the accumulation of T-bet⁺ ASCs. Finally, we show that the nascent reengagement of memory B cells in the extrafollicular pathway is associated with post-RTX relapses. In summary, we pinpoint the extrafollicular B cell response as a possible origin for autoreactive ASCs in childhood INS.

## Results

### Perturbation of the B cell transcriptional landscape is the major immunological abnormality in childhood INS

Since the transcriptional landscape of immune cells in pediatric INS had not yet been defined, we aimed to characterize the nephrotic immune signature by scRNA-seq. Following doublet and non-viable cell removal in standard, quality control steps (Supplementary Fig. 1a–c), we analyzed the transcriptomes of 69,994 immune cells in PBMC of four children during active INS without known viral infection (INS; 32,139 cells) and four age/sex-matched HCs (Supplementary Data 1). Following integrated clustering, we identified 18 distinct immune cell populations that were uniformly present in all donors (Fig. 1a and Supplementary Fig. 2a, b). Cluster identities were determined by the expression of canonical lineage-defining genes and were

subsequently stratified into broad immune cell lineages: B cells (expression of *CD19*, *CD79A*, *CD79B*), CD4⁺ T cells (*CD3G*, *CD4*), CD8⁺ T cells (*CD3G*, *CD8*), double negative T cells (*CD3G* and lacking *CD4* and *CD8*), NKT cells (*CD3G*, *KLRB1*, *KLRG1*), NK cells (*ZBTB16*, *NKG7*, *GNLY*), monocytes/dendritic cells (*CD14* or *FCGR3A*), and plasmacytoid dendritic cells (*LILRA4*) (Fig. 1a and Supplementary Fig. 2c, d). Notably, the only lineage that was preferentially expanded in children with INS was the B cell lineage (Fig. 1b). Specifically, the memory B cell-containing clusters C11 and C12 and the antibody-secreting cell (ASC) cluster C13 accumulated in INS indicating the induction of a humoral response (Supplementary Fig. 2b). A cluster of NK cells relating to CD16^dim (C9) was the only other cluster expanded in INS.

To identify a transcriptional profile associated with INS, we performed pseudobulk differential gene expression analysis between INS and HC children for each broad immune cell lineage[37]. Through this pseudobulk approach, cells were aggregated at the level of the donor and broad immune cell lineages to account for biological replication. We identified 1976 genes that were differentially expressed ($|\log_2 FC| > 0.65$, $P_{adj} < 0.05$) in at least one lineage and present in at least 10% of cells of that lineage (Fig. 1c, Supplementary Fig. 2e, and Supplementary Data 2). The largest transcriptional differences were amongst B cells encompassing 958 genes, 642 of which were upregulated in INS (Fig. 1c, d and Supplementary Fig. 2e). We defined the nephrotic B cell signature as these 642 upregulated genes (Fig. 1d, Supplementary Data 2). Hence, perturbations in the B cell transcriptional landscape represents the major immune abnormality in the blood during active childhood INS.

### B cells in INS are poised for acquisition of effector functions and ASC differentiation

Pathogenic B cells possess both antibody-dependent and -independent functions that can trigger and drive autoimmunity[21]. To assess the functional properties of B cells in INS, we performed pathway analysis in the nephrotic B cell signature and found a substantial enrichment of terms associated with the engagement of humoral immunity (Fig. 1e). INS B cells had elevated expression of genes encoding components of B cell receptor (BCR) signaling including the tyrosine kinases *SYK* and *BTK*, adapters *BLNK*, *BANK1*, and *LAT2*, and the co-receptor *CD19* denoting an activated status (Fig. 1f). This activated phenotype was further supported by the increased expression of activation-associated genes like the APRIL/BAFF receptor TACI (*TNFRSF13B*), the memory marker *CD27*, and both chains of the activating integrin VLA-4 (*ITGA4*, *ITGB1*) (Fig. 1f). Moreover, the transcriptional landscape of INS B cells revealed the acquisition of key effector functions including immunoglobulin production as evidenced by the elevated expression of several variable heavy and light chain genes, *IGHG1*, *IGHG3*, *IGHA2*, and *IGHA1* (Fig. 1f). The molecular chaperone *MZB1* that mediates IgM and IgA secretion in marginal zone (MZ) B cells, B-1 cells, and ASCs was particularly enriched in INS B cells (Fig. 1f). Oxidative phosphorylation (OXPHOS) and fatty acid oxidation (FAO) pathways were also elevated in INS B cells consistent with a metabolic program essential for ASC development and antibody generation (Fig. 1e, f)[38,39]. Beyond antibody generation, INS B cells upregulated the expression of genes involved in antigen presentation, including *HLA-DPA1* and *HLA-DOB*, the cathepsin *CTSS*, and the lipid antigen presenter *CD1C* (Fig. 1e, f). We also observed an enrichment of genes involved in actin cytoskeleton dynamics including the Arp2/3 complex (*APRC1B*, *ARPC5*), actin nucleation factors (*WAS*), polymerization factors (*EVL*, *VASP*), and capping proteins (*CAPG*, *CAPZB*) highlighting increased B cell motility in INS (Fig. 1e, f)[40]. This enrichment of BCR signaling, B cell activation, antigen presentation, actin polymerization, and fatty acid oxidation pathways in INS B cells was also observed by gene set enrichment analysis (GSEA) (Supplementary Fig. 3a).

Next, we sought to identify the putative transcriptional drivers of the nephrotic B cell signature using the web-based transcription factor

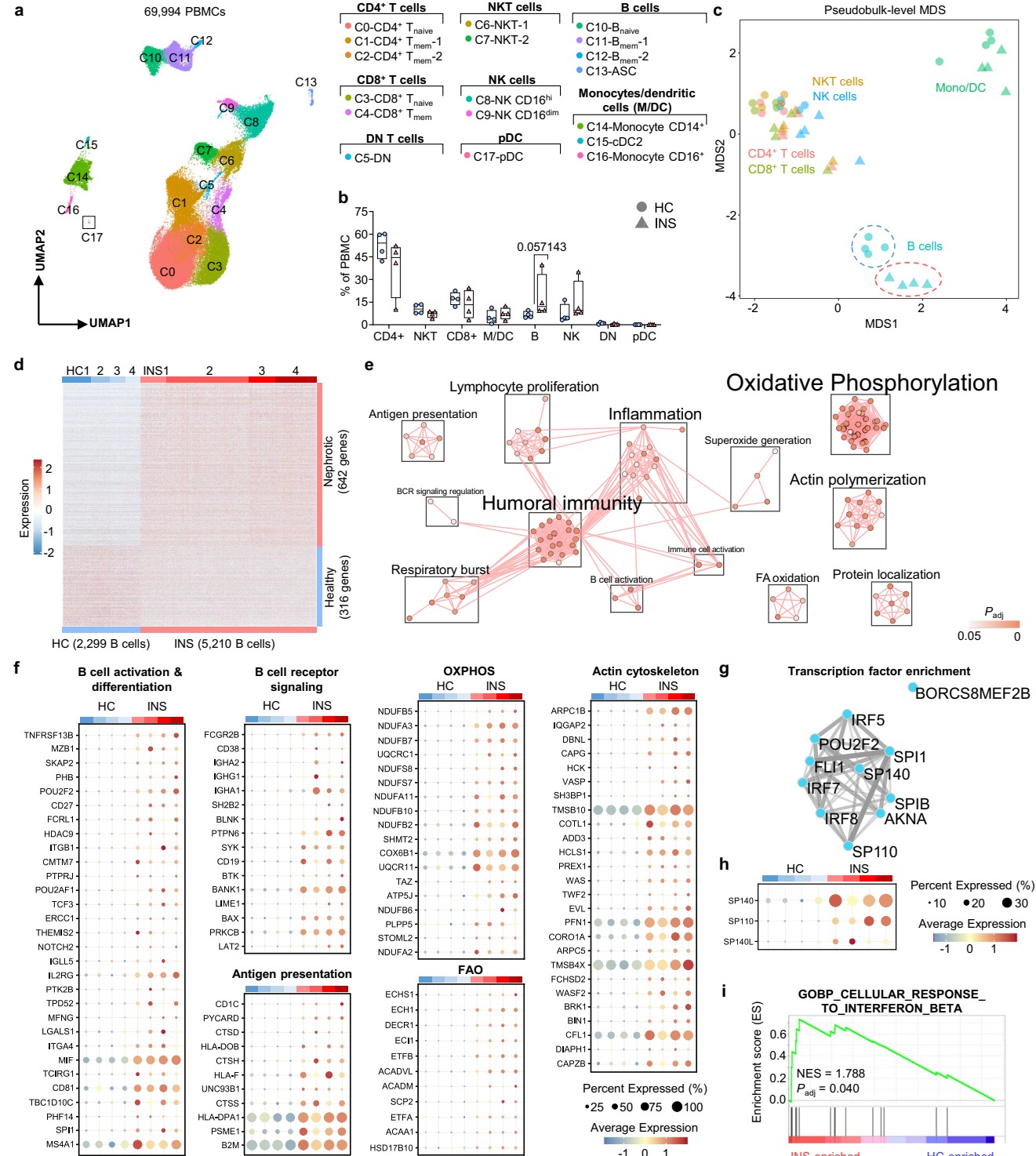

**Fig. 1 | B cells in INS are transcriptionally poised to acquire effector functions.**
**a** Integrated Uniform Manifold Approximation and Projection (UMAP) of the 18 clusters of PBMC from HC ($n = 4$) and INS ($n = 4$) children. **b** Proportions of each broad immune cell lineage. Data are shown as box plots depicting the median (center), interquartile range (bounds of box), and min-max range (whiskers); each dot corresponds to a single donor ($n = 4$ HC, 4 INS); $P$-values were determined using individual two-sided Mann-Whitney $U$-tests. **c** Pseudobulk differential gene expression analysis between INS and HC broad cell lineages was performed with the Muscat R package using the edgeR method and a multi-dimensional scaling (MDS) plot was used to depict the results. Genes with |log₂(Fold Change)| > 0.65 and $P_{adj}$ < 0.05 were considered significantly differentially expressed. $P_{adj}$ were determined using the Benjamini-Hochberg correction **d** Heatmap of the normalized

expression of differentially expressed genes between INS and HC B cells.
**e** Enrichment map depicting the pathway analysis results in the nephrotic B cell signature. **f** Bubble plots showing the expression of genes associated with B cell effector functions in the nephrotic B cell signature. OXPHOS denotes oxidative phosphorylation and FAO denotes fatty acid oxidation. **g** A network plot from ChEA3 depicting the transcription factors predicted to confer the nephrotic B cell signature. **h** A bubble plot showing the expression of three ChEA3 hits. **i** A gene set enrichment analysis (GSEA) plot of the Gene Ontology term "Cellular Response to Interferon Beta" (GO:0035458). $P_{adj}$ value was determined using the Benjamini-Hochberg correction for multiple-testing. NES, normalized enrichment score.

enrichment analysis tool ChIP-X Enrichment Analysis 3 (ChEA3)[41]. Transcriptional targets for PU.1 (*SPI1*), SPI-B (*SPIB*), and OCT-2 (*POU2F2*), key transcription factors coordinating B cell activation and functionalization, were enriched in INS B cells (Fig. 1g and Supplementary Data 4)[42]. Consistently, the genes encoding these transcription factors were themselves upregulated (Fig. 1f). Collectively, these data denote that B cells in INS are activated and can exert potentially pathogenic effector functions.

We also observed possible transcriptional regulation by the speckled protein (SP) chromatin readers SP140, SP110, and SP140L, the expression of which were also strongly elevated in INS B cells (Fig. 1g, h). These transcription factors were recently implicated in antiviral type-I interferon (IFN) responses, though their function in B cells remain undefined[43,44]. Interestingly, we identified a type-I IFN signature in INS B cells conferred by genes downstream IFN-β signaling (e.g., *IFNAR2*, *OAS1*, *AIM2*, *IFITM2*, *IFITM3*, *XAF1*, *PYHIN1*, *MNDA*, *CAPN2*, *IKBKE*) (Fig. 1i and Supplementary Fig. 3b). Most of these genes were expressed in <10% of B cells and were thus not included in the nephrotic B cell signature. Nevertheless, these results indicate that type-I IFN signaling, a common driver of antiviral immunity and autoimmunity, may underline the generation of the nephrotic B cell response.

## Childhood INS is characterized by the expansion of extra-follicular B cell populations

Having demonstrated the increased expression of genes associated with B cell activation and function in children with active INS, we sought to identify the pathogenic B cell subsets underlying the nephrotic B cell response. Integrated subclustering of the B cell lineage identified ten distinct subpopulations corresponding with naïve B cells (*BACH2*-expressing subclusters B0, B4, and B7), memory B cells (*BCL2A1*-expressing subclusters B1, B2, B3, B5, and B6), and ASCs (*PRDM1*-expressing subclusters B8 and B9) (Fig. 2a, b, Supplementary Fig. 4a and Supplementary Data 5). The naïve clusters included transitional naïve B cells defined by *IGHM*, *NEIL1*, *HRK*, and *TCL1A* expression (subcluster B4), and mature naïve B cells defined by *IGHM*, *FCER2*, and *IL4R* expression (subclusters B0 and B7) (Fig. 2b and Supplementary Fig. 4b). Subcluster B0, representing the largest naïve B cell subcluster in all children, was significantly reduced in INS while two memory (B3 and B6) and all ASC subclusters (B8 and B9) were increased indicating elevated antigen-experience in INS (Fig. 2c).

To define the memory B cell subclusters, we performed differential gene expression analysis between each memory B cell subcluster identified (Fig. 2d). Cells in subcluster B1 expressed genes associated with activation (*CD69*, *FOS*, and *FOSB*) and IgM and IgD synthesis (*IGHM* and *IGHD*) denoting an isotype-unswitched phenotype and were accordingly termed activated memory B cells (actMBC) (Fig. 2d). Cells in the dominant, INS-associated subcluster B3 were also isotype-unswitched and expressed genes associated with extrafollicular MZ B cells (*CD1C*, *CD24*, *PLD4*)[45,46], and were thus termed MZ-like B cells (Fig. 2d). Consistent with an extrafollicular phenotype, MZ-like B cells also preferentially expressed *TNFRSF13B* (TACI), the APRIL/BAFF receptor that drives extrafollicular ASC generation[22], and *GPR183* (EBI2), the G protein-coupled receptor that orchestrates extrafollicular reactions by homing B cells to extrafollicular foci[47]. Cells in subcluster B5 corresponded with isotype-switched memory (SM) B cells as they lacked *IGHM* and *IGHD* expression and instead expressed *IGHG1*, *IGHA1*, and *IGHA2* (Fig. 2d). Both the MZ-like B cell and SM subpopulations exhibited a transcriptional profile consistent with memory B cells including *ANXA2*, *S100A10*, and *S100A4*. Subcluster B2, termed MBC-2, was similar to MZ-like and SM B cell subpopulations, although the extent of the memory B cell phenotype was diminished (Fig. 2d). Finally, the second INS-associated subcluster B6 corresponded with atBCs, a B cell population that participates in extrafollicular B cell responses arising in autoimmune and chronic viral

infection settings[27,28,33]. Genes characteristic of atBCs were enriched in this subcluster, including *FCRL5*, *FCLRA*, *FCRL2*, *ITGB2*, *ITGB7*, *ITGAX* (CD11c), *FGR*, *ZEB2*, *NR4A2*, *ZBTB32*, and high *CD19* expression (Fig. 2d and Supplementary Fig. 4c). Of the immunoglobulin heavy chain genes, atBCs preferentially expressed *IGHD* and *IGHM* denoting reduced class-switching thereby supporting an extrafollicular origin (Fig. 2d). Interestingly, the two subclusters expanded in INS, namely MZ-like B cells and atBCs, preferentially expressed *POU2F2*, one of the putative transcriptional drivers of the nephrotic B cell signature (Fig. 2d).

Pseudobulk differential gene expression analysis showed significant upregulation of genes in all the B cell subclusters in INS, except for atBCs and ASCs (Fig. 2e and Supplementary Fig. 5a). Pathway analysis revealed an enrichment of functions associated with actin cytoskeletal dynamics in all B cell subclusters in INS, while some subclusters (Naïve-1, actMBCs, MBC-2, and MZ-like B cells) also showed an enrichment of B cell activation pathways (Supplementary Fig. 5b). The antigen-inexperienced and recently activated B cell subsets (Naïve-1 and actMBC) showed the greatest differences in gene expression suggesting that the transcriptional events leading to B cell dysregulation occur early during B cell activation (Fig. 2e and Supplementary Fig. 5a). There was a substantial overlap in the upregulated genes between Naïve-1 B cells, actMBCs, MBC-2, and MZ-like B cells including many atBC-associated genes (e.g., *ITGB2*, *ITGB7*, *FCRLA*, *FCRL2*), *POU2F2*, *POU2AF1*, and the extrafollicular response genes *TNFRSF13B* and *GPR183*, suggesting a possible developmental sequence or cooperation between these B cell subclusters in generating an extrafollicular B cell response in INS (Fig. 2e).

We sought to better identify relationships between the B cell subclusters by performing trajectory inference using Monocle3, a computational method that orders cells along a pseudotemporal trajectory informed by changes in gene expression to make inferences about cell development[48]. To this end, ASCs were removed from analysis and the remaining B cells were re-clustered before selecting the transitional naïve B cell cluster as the starting point for calculating pseudotime (Fig. 2f). SM B cells and atBCs formed the termini of two independent branches (II and III) that arose from a common branching point in the MBC-2 subcluster. Branch III was comprised of the INS-expanded MZ-like B cells and atBCs suggesting an origin for atBCs in MZ-like B cells, consistent with a recent study on atBCs from malaria-infected adults[27]. As both atBCs and MZ-like B cells are extrafollicular B cell populations, trajectory III may represent an extrafollicular developmental pathway for B cells. Indeed, the expression of both *TNFRSF13B* and *GPR183* was higher in trajectory III than trajectory II, with *TNFRSF13B* being only expressed in B cells from INS children (Fig. 2f).

Monocyte-derived dendritic cells within secondary lymphoid organs are established sources of TACI ligands (APRIL and BAFF) during extrafollicular B cell responses[49,50]. To identify potential sources of these TACI ligands in PBMC of INS children, we evaluated the APRIL and BAFF signaling networks by CellChatDB[51]. The APRIL signaling network consisting of APRIL (*TNFSF13*), TACI (*TNFRSF13B*), and BCMA (*TNFRSF17*) was specifically enriched in PBMC from INS children, with monocytes and dendritic cells being the predominant sources of APRIL (Fig. 2g, h). Accordingly, both ASC subclusters (subcluster B8 and B9) were identified as the predominant receivers of APRIL signal (Fig. 2g), and both were highly expanded in INS children blood (Fig. 2a, c). Collectively, these data demonstrate that memory B cells in childhood INS are engaged through an extrafollicular pathway with a capacity to generate ASCs.

## Proliferating T-bet⁺ atBCs and ASCs are a hallmark of active INS

The rapid extrafollicular expansion of ASCs is recognized as a salient feature of antiviral immunity and has only recently since been linked to multiple autoimmune conditions[25,27–29,31,52]. Since we identified a

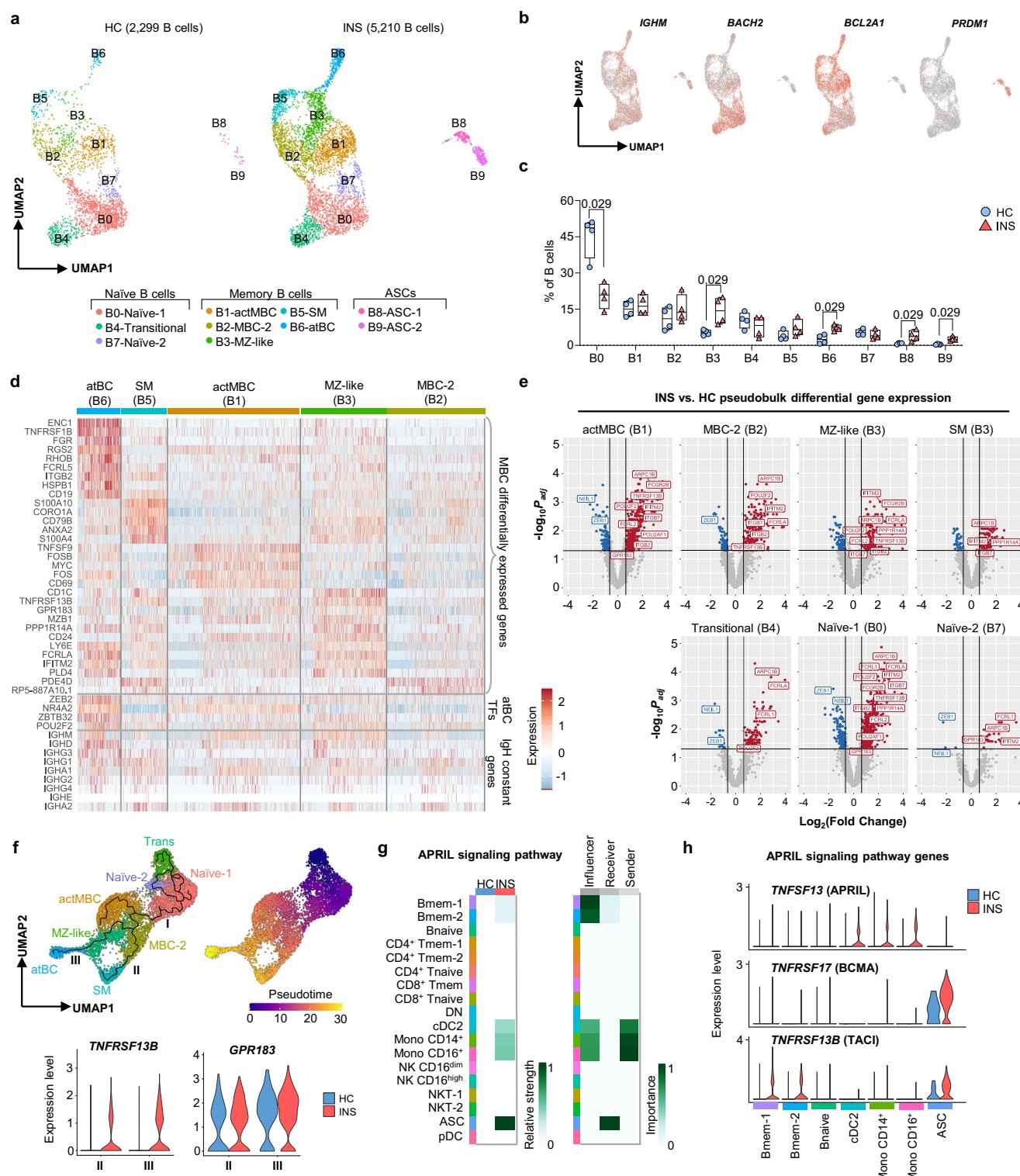

pronounced extrafollicular mobilization of memory B cells in INS, we sought to confirm the presence of extrafollicular B cells using the most up-to-date flow cytometric definitions for these cells in the peripheral blood of 13 children with active INS and 24 age-matched HCs. Total CD19[+] B cell frequencies did not differ between HC and INS children (Supplementary Fig. 6a). Following unsupervised high-dimensional clustering of B cells, we generated 14 distinct B cell metaclusters (M0-M13) present in HC and INS children (Fig. 3a, b). The metaclusters were broadly categorized into naïve (CD27[−]CD21[+]IgM[+]IgD[+]), classical memory (cMBC: CD27[+]CD21[+]CD20[+]), CD21[low] memory

(CD27[+/−]CD21[low]CD20[+]CD38[−/low]), and ASC (CD20[−]) populations (Fig. 3b, Supplementary Fig. 6b). INS children had lower frequencies of the largest naïve B cell metacluster (M0) and an expansion of an isotype-switched (IgM[−] IgD[−]) cMBC metacluster (M9), consistent with previous reports[4,10,53]. However, most of the metaclusters that were expanded in INS corresponded with isotype-switched CD21[low] memory B cells (M7, M11) and both isotype-unswitched (M13) and switched (M8, M12) CD38[high] ASCs (Fig. 3c–e). The INS-expanded CD21[low] metacluster M11 and all ASC metaclusters (M8, M12, M13) were Ki-67[+] denoting active cycling, and the proportions of Ki-67[+] B cells were

**Fig. 2 | The engagement of extrafollicular B cells is a feature of active INS.**
**a** Integrated UMAP of the ten B cell subclusters shown for HC (n = 4) and INS (n = 4).
**b** Feature plots showing expression of *IGHM*, *BACH2*, *BCL2A1*, and *PRDM1* in B cells.
**c** Proportions of each B cell subcluster in HC and INS children. Data are shown as box plots depicting median (center), interquartile range (bounds of box), and min-max range (whiskers); each dot corresponds to a single donor (n = 4 HC, 4 INS); P-values were determined using individual two-sided Mann-Whitney U-tests.
**d** Heatmap showing the expression of the most highly enriched genes in each memory B cell subcluster. **e** Pseudobulk differential gene expression analysis between INS and HC B cell subclusters was performed with the Muscat R package using the edgeR method and volcano plots were used to depict the results. Genes with |log₂(Fold Change)| > 0.65 and $P_{adj}$ < 0.05 were considered significantly differentially expressed. $P_{adj}$ values were determined using the Benjamini-Hochberg

correction **f** UMAP plots showing re-clustering of all B cell subclusters except ASC-1 and ASC-2 alongside the trajectory (left) and pseudotime values (right) determined by Monocle3, and violin plots showing the expression of *TNFRSF13B* and *GPR183* in branches II and III (bottom). **g** Heatmaps from CellChatDB showing the relative strength of the participation of each PBMC cluster in the APRIL signaling network in HC and INS (left), and the importance of each PBMC cluster in INS as acting as "sender", "modulator", or "receiver" cell types within the APRIL signaling network. **h** Violin plots showing the expression of genes within the APRIL signaling network across participating PBMC clusters in HC and INS. actMBC, activated memory B cells; MBC-2, memory B cell cluster 2; MZ-like, marginal zone-like B cells; SM, isotype-switched memory B cells; atBC, atypical B cells; ASC, antibody-secreting cells.

---

significantly greater in INS compared to HC (Fig. 3d, e). Moreover, both INS-expanded CD21^low populations had diminished levels of CXCR5, the key chemokine receptor orchestrating germinal center responses, denoting an extrafollicular origin for these B cells (Fig. 3d, e). In contrast, the INS-expanded isotype-switched cMBC metacluster was CXCR5^high Ki-67^- denoting a follicular origin (Fig. 3d, e).

The expansion of CD21^low B cells is an established feature of B cell dysregulation and frequently relates with atBC responses in autoimmunity and viral infections[31,54–57]. We sought to determine whether the expansion of CD21^low B cells in INS correlated with the increased frequency of atBCs identified by scRNA-seq. T-bet^+ CD11c^+ B cells, one of the most widely accepted phenotypes for atBCs[30,32,58], were almost exclusively present within the CD21^low B cell (CD27^+/-CD21^lowCD20^+CD38^-/low) compartment representing ~40% of the cells (Fig. 4a, Supplementary Fig. 7a). Accordingly, frequencies of T-bet^+ CD11c^+ B cells strongly correlated with frequencies of CD21^low B cells (Fig. 4a). Indeed, atBCs (T-bet^+ CD11c^+ CD27^+/- CD21^low CD20^+ CD38^-/low), and more specifically the isotype-switched atBCs, were significantly expanded in INS (Fig. 4b). In line with previous reports and our scRNA-seq data, these atBCs expressed higher levels of CD19 than cMBCs (Supplementary Fig. 7b). Thus, the accumulation of CD21^low B cells in INS corresponds with an expansion of atBCs. IgD^- CD27^- double negative (DN) B cells, another common definition for atBCs[24,25], were also increased in INS (Supplementary Fig. 7c).

To identify differences in INS and HC atBCs, we compared the expression of T-bet, a key driver of the atBC transcriptional program essential for their conversion into ASCs, and FcRL5, an inhibitory receptor that restricts BCR signaling[59]. atBCs in INS expressed higher levels of T-bet and lower levels of FcRL5 indicating a stronger tendency to develop into ASCs (Fig. 4c, d). Consistently, ASCs (CD20^- CD10^- CD38^high) were greatly expanded in INS, with a greater proportion accumulating as isotype non-switched ASCs suggesting an extrafollicular origin (Fig. 4e). As with atBCs, ASCs from INS children had greater T-bet expression and were largely lacking FcRL5 (Fig. 4f, g). In summary, the expansion of isotype-switched atBCs and T-bet^+ ASCs in active INS positions the extrafollicular B cell response as a hallmark of active disease.

## RTX effectively ablates INS-associated B cell populations

While relapses are effectively controlled using GCs, a more enduring remission state is achieved with subsequent RTX therapy despite the eventual resurgence of peripheral B cells[6]. As such, we aimed to compare the impact of GC alone and GC/RTX combination therapy on the INS-associated B cell populations (Fig. 5a, Supplementary Table 1 and Supplementary Data 1). We first compared B cell numbers in the blood of actively relapsing individuals (INS) to those in GC-induced remission (Rem-GC) and remission maintained by RTX (Rem-RTX). Treatment with GC strongly decreased peripheral ASC numbers with no major impact on earlier B cell subpopulations indicating a limited effect of GCs on the INS-associated B cell response (Fig. 5b–e). In contrast, children maintained in remission with RTX had lower total B

cell numbers than RTX-inexperienced individuals, suggesting incomplete B cell recovery at the time of sampling post-RTX (Fig. 5c). At this time point, children had reconstituted the transitional naïve B cell compartment, but cMBCs, atBCs, and ASCs remained sparse in peripheral blood (Fig. 5b, d, e). Hence, while GC treatment was effective at restricting ASC abundance in the blood, a more extensive B cell deficiency that eliminated all INS-associated B cell populations was achieved with subsequent RTX administration.

Next, we aimed to evaluate the changes in the B cell compartment in relapses following remission maintained by RTX. B cell numbers were still largely suppressed in patients in post-RTX relapse (Supplementary Fig. 8a–c). Thus, we predicted that changes in B cell subset composition rather than total cell numbers may differentiate between post-RTX remission (Rem-RTX) and relapse (Rel-RTX). While frequencies of total B cells in PBMC were similar in both groups (Fig. 5f), those who relapsed had fewer proportions of transitional naïve B cells (confirmed in 3/5 longitudinal samples) and higher proportions of both isotype-unswitched and switched cMBC (confirmed in 5/5 longitudinal samples), confirming an earlier finding that the resurgence of isotype-switched cMBCs is associated with post-RTX relapses of INS (Fig. 5b, g, h)[20]. Surprisingly, we did not observe any differences in atBC and ASC frequencies in post-RTX remission and relapse (Fig. 5b, h).

## The nascent resurgence of MZ-like B cells is associated with post-RTX relapse

Given that the B cell compartment in post-RTX relapses was still largely skewed towards antigen-inexperienced populations, we hypothesized that this represented an early time point in the generation of the nephrotic B cell response. To investigate this, we performed scRNA-seq on peripheral B cells isolated from four of the children from whom we were able to obtain both relapse and remission samples following RTX treatment (Supplementary Data 1). Following integrated clustering, we obtained 12 clusters most of which represented antigen-inexperienced cells as is evidenced by *BACH2* expression (Fig. 6a, b, Supplementary Fig. 9a). Three clusters (clusters R4, R9, R10) expressed *BCL2A1* and corresponded to MZ-like B cells, SM B cells, and atBC subpopulations (Fig. 6a, b, Supplementary Fig. 9a). Only MZ-like B cells (cluster R4), the predominant B cell subset associated with active INS, were expanded in all relapsing individuals supporting an early resurgence of the nephrotic B cell response (Fig. 6c). Accordingly, the MZ-like B cell signature (3/4 children), and to a lesser extent the atBC signature (2/4 children), was enriched in B cells during post-RTX relapse (Fig. 6d, Supplementary Fig. 9b). This expansion of MZ-like B cells reflects a nascent resurgence of the extrafollicular B cell response in post-RTX relapse.

Finally, we sought to evaluate the similarity between this nascent extrafollicular response, and the B cell responses observed in active INS. Pseudobulk differential gene expression analysis did not yield significant results, likely due to the overwhelmingly naïve B cell landscape in the post-RTX setting. As such, we proceeded with a non-pseudobulk approach comparing gene expression in all memory B

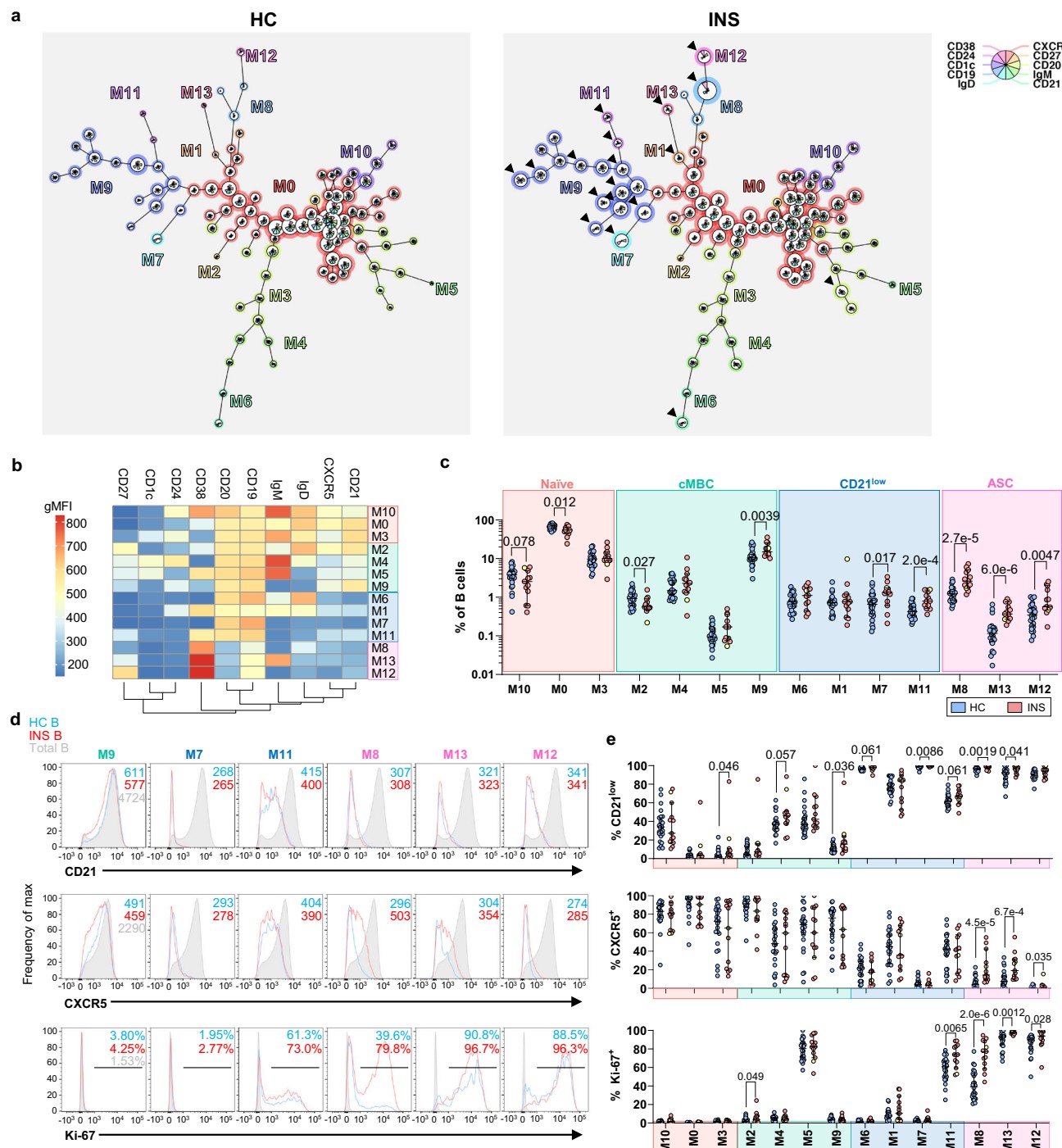

**Fig. 3 | CD21^low CXCR5^− B cells and ASCs are actively expanding in active INS.**
**a** Minimal spanning trees (MST) of HC (*n* = 24) and INS (*n* = 13) B cells generated by
FlowSOM clustering. The 14 metaclusters (M0-M13) are labeled on both MSTs.
Arrowheads indicate key cell clusters expanded in INS. **b** Heatmap showing the
geometric mean fluorescence intensities (gMFI) of the markers used for clustering
in each metacluster. **c** Proportions of each metacluster in HC (*n* = 24) and INS
(*n* = 13) PBMC. **d**, **e** Histograms of CD21, CXCR5, and Ki-67 in the six metaclusters

enriched in INS (**d**) with quantification of the proportions of CD21^low, CXCR5^+, and
Ki-67^+ cells in each metacluster from HC (*n* = 24) and INS (*n* = 13) B cells (**e**). Data are
shown as median with 95% confidence intervals and *P*-values were determined by
two-sided Mann-Whitney *U*-tests in (**c–e**). Each data point corresponds to a single
donor. The yellow data point represents the child with glucocorticoid-resistant
membranous nephropathy. cMBC, classical memory B cells; ASC, antibody-
secreting cells.

cells (clusters R4, R9, and R10) between relapse and remission time-
points. We identified small differences in gene expression represented
by log$_2$FC magnitudes of <0.2 (Fig. 6e, Supplementary Data 6). The
most enriched genes (|log$_2$FC| > 0.1, $P_{adj}$ < 0.01) in post-RTX relapse
memory B cells were present within the nephrotic B cell signature
(Fig. 6f). These included genes associated with MZ-like B cells and

atBCs (*POU2F2, CD19, PPP1R14A, COTL1, LY6E*, and *RGS2*), as well as
genes involved in the type-I IFN response (Fig. 6f). Consistently, the
response to IFN-β signaling GO term was highly enriched in post-RTX
relapse memory B cells (Supplementary Fig. 9c). Similar findings were
observed in the naïve compartment (clusters R0, R1, R2, R3, R5, R6, R7,
and R8) during post-RTX relapse (Supplementary Fig. 10a). Here, there

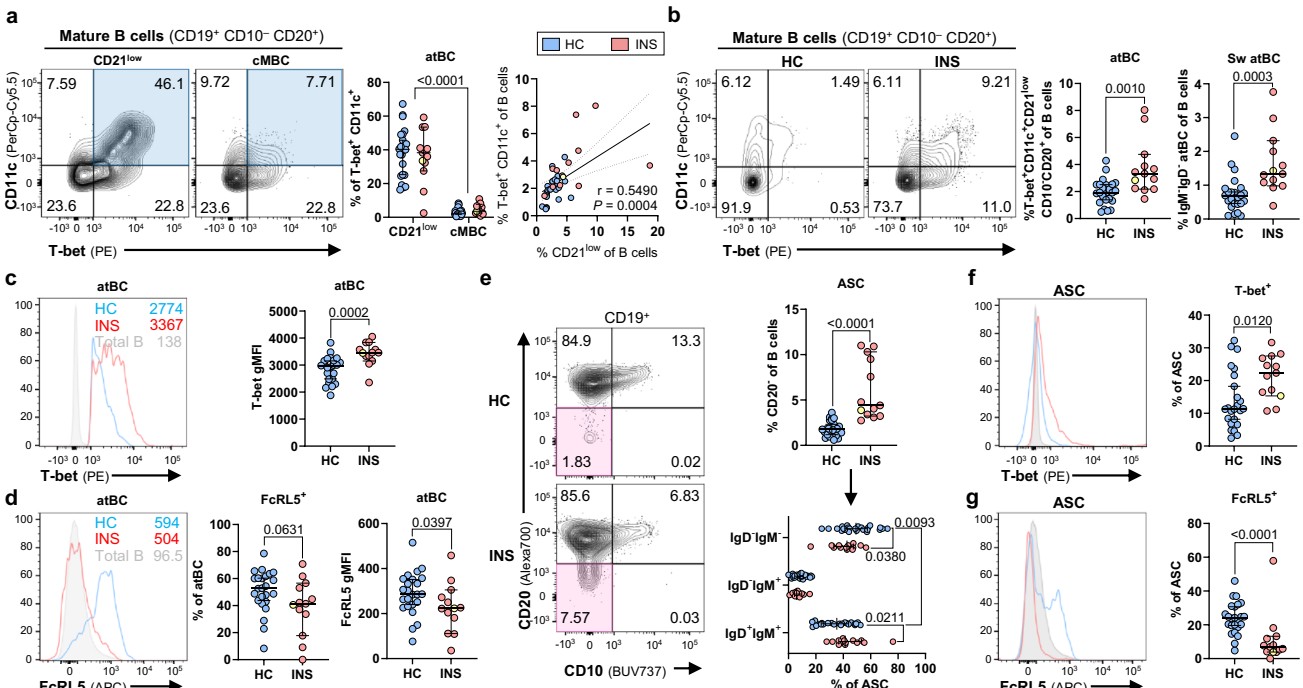

**Fig. 4 | CD21$^{low}$ T-bet$^+$ atBCs and ASCs are a hallmark of childhood INS.**
**a** Representative flow plots highlighting T-bet$^+$CD11c$^+$ cells (*aquamarine*) in the CD21$^{low}$ (CD21$^{low}$ CD19$^+$ CD10$^-$ CD20$^+$ CD38$^{-/low}$) and cMBC (CD27$^+$ CD21$^+$ CD19$^+$ CD10$^-$ CD20$^+$ CD38$^{-/low}$) mature B cell compartments. Proportions of T-bet$^+$ CD11c$^+$ among CD21$^{low}$ and cMBC mature B cells (*left graph*) in HC (*n* = 24) and INS (*n* = 13) along with linear regression analysis between the proportions of T-bet$^+$ CD11c$^+$ and CD21$^{low}$ mature B cells (*right graph*). **b** Representative flow plots of atBCs (T-bet$^+$ CD11c$^+$ mature B cells) in HC and INS. Quantification of the proportion of atBCs (*left graph*) and isotype-switched (IgM$^-$ IgD$^-$) atBCs (*right graph*) in HC (*n* = 24) and INS (*n* = 13) B cells. **c**, **d** Histograms showing T-bet (**c**) and FcRL5 (**d**) expression in HC (*n* = 24) and INS (*n* = 13) atBC with quantification of MFIs and cell frequencies.

**e** Representative flow plots highlighting ASCs (*pink*) in total B cells. Quantification of the proportions of ASCs in HC (*n* = 24) and INS (*n* = 13) B cells (*top right graph*) and their isotype distributions (*bottom right graph*). **f, g** Histograms showing T-bet (**f**) and FcRL5 (**g**) expression in HC (*n* = 24) and INS (*n* = 13) ASCs with quantification of cell frequencies. Data are shown as median with 95% confidence intervals and *P*-values were determined by two-sided Mann-Whitney *U*-tests (**b**–**g**) two-way ANOVA with Tukey's multiple testing ((**a**) *left*, (**e**) *bottom*), or Pearson's correlation ((**a**) right). Each data point corresponds to a single donor. The yellow data point represents the child with glucocorticoid-resistant membranous nephropathy. cMBC classical memory B cells, atBC atypical B cells, ASC antibody-secreting cells.

was an enrichment of GO terms associated with B cell activation and type-I IFN signaling (Supplementary Fig. 10b). These results support the early re-establishment of the extrafollicular nephrotic B cell response during post-RTX relapse.

## Discussion

A B cell origin for the pathogenesis of childhood INS is supported by the efficacy of B cell-depleting biologics like RTX at maintaining long-term remission from proteinuria. Nevertheless, the precise nature of the pathogenic B cell response has thus far remained elusive. In this study, we used scRNA-seq and multi-parametric flow cytometry to identify a dysregulated B cell response and transcriptional signature associated with active INS. This signature was conferred by the expansion of CD21$^{low}$ CXCR5$^-$ CD11c$^+$ T-bet$^+$ atBCs—a B cell population that is associated with extrafollicular responses in chronic viral infection and systemic autoimmunity—and T-bet$^+$ ASCs. Moreover, we demonstrated that distinct immunosuppressive treatment strategies, namely GC and RTX, differentially targeted these INS-associated B cell populations with RTX providing more extensive coverage. Altogether, our study uncovers prominent involvement of an extrafollicular B cell response in pediatric INS.

Using scRNA-seq on total PBMC isolated from INS and HC children, we showed that perturbations in the B cell compartment represented the major immunological abnormality in pediatric INS. In comparison to healthy B cells, INS B cells upregulated genes involved in BCR signaling, antibody production, antigen presentation, oxidative phosphorylation/fatty acid oxidation, and actin cytoskeleton

dynamics. This transcriptional signature was possibly endowed by PU.1 (*SPI1*), SPI-B (*SPIB*), OCT-2 (*POU2F2*), and OBF-1 (*POU2AF1*), transcription factors that coordinate the expression of multiple BCR signal transducers and receptors enabling B-T cell communication, and thereby take a central position in B cell acquisition of effector functions[42]. These data indicate that B cells receive an activating signal and acquiring effector functions during active disease. The nephrotic signature was present in all four children with active INS despite participants being in very different stages of active disease (first episode before therapy, relapse before therapy, relapse on therapy) possibly denoting a uniform mechanism for B cell engagement at the time of active disease.

By further stratifying the B cell compartment into distinct sub-populations, we observed that INS patients had a reduction in naïve B cells and an expansion in MZ-like B cells, atBCs, and ASCs. MZ-like B cells represented an isotype-unswitched memory B cell subset that was almost exclusively present in INS children, and expressed genes associated with MZ B cells (*IGHM, IGHD, CD24, CD1C, PLD4*), an extra-follicular B cell population that largely resides in the spleen giving rise to IgM-secreting ASCs in a T cell-independent manner[60,61]. The atBCs expanded in INS were equivalent to those recently identified in people with viral or parasitic infections (malaria, HIV, SARS-CoV-2), auto-immunity (SLE, rheumatoid arthritis, and multiple sclerosis), and immunodeficiencies (CVID, partial RAG deficiency)[25,27,28,31–33]. Recent reports demonstrate that this population is involved in extrafollicular B cell responses acting as an important source for short-lived ASCs. Thus, the expansion of both MZ-like B cells and atBCs is strongly

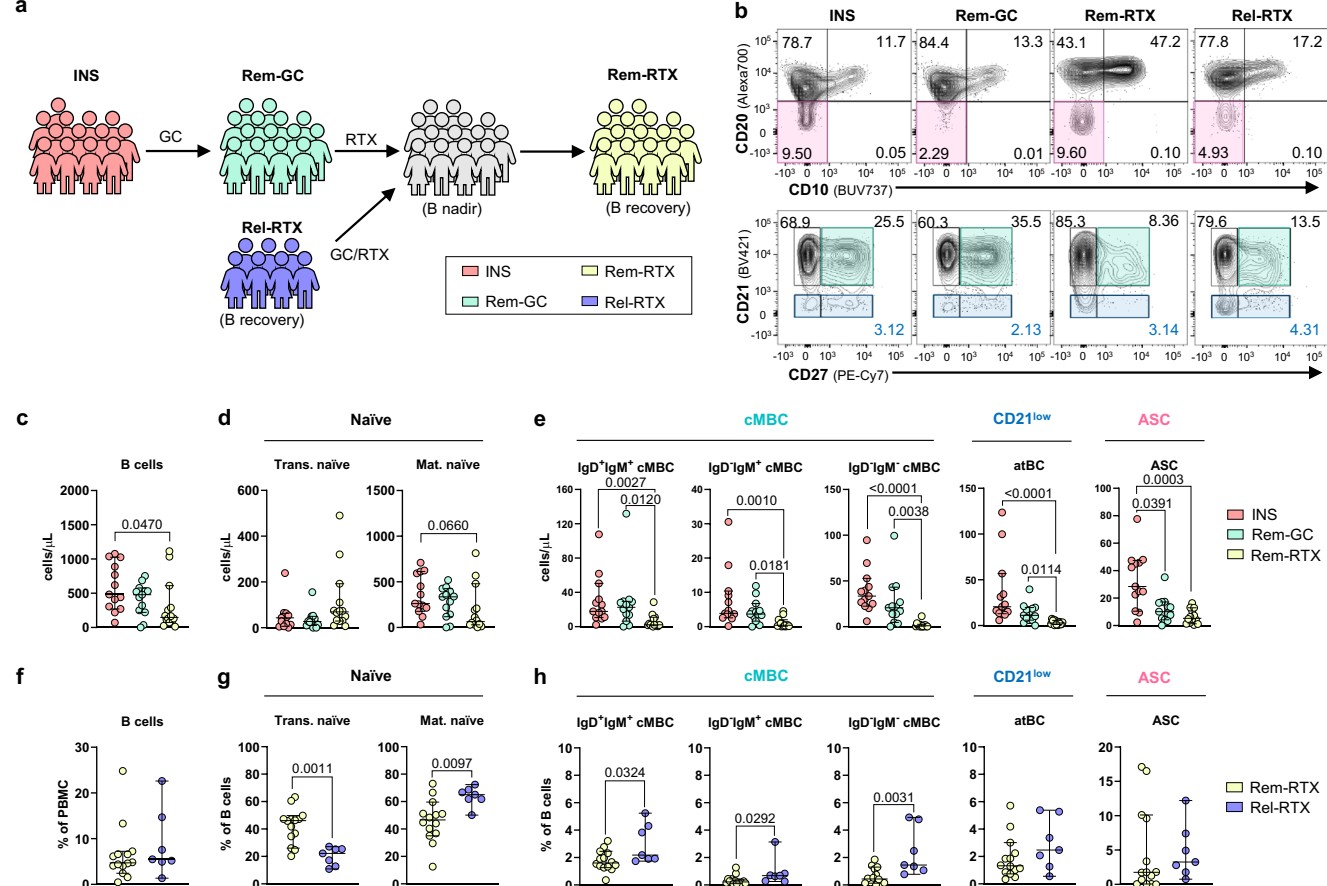

**Fig. 5 | atBCs and ASCs are effectively targeted by RTX. a** Schematic representing the patient groups from which PBMC were obtained in the observational longitudinal cohort. Rem-GC, remission on glucocorticoids; Rem-RTX, remission maintained by rituximab; Rel-RTX, relapse following rituximab (**b**) Representative flow plots from each patient group showing total B cells (CD19⁺, *top*), and mature B cells (CD19⁺ CD10⁻ CD20⁺, *bottom*). Colored boxes denote ASCs (*pink*), cMBC (*teal*), and CD21^low atBCs (*aquamarine*). **c**–**e** Absolute numbers of total (**c**), naïve (**d**), and memory (**e**) B cell populations in the blood of INS (*n* = 13), Rem-GC (*n* = 14), and

Rem-RTX (*n* = 14) individuals. **f**–**h** Proportions of total (**f**) naïve (**g**) and memory (**h**) B cells in the blood of Rem-RTX (*n* = 14) and Rel-RTX (*n* = 7) individuals. Data are shown as median with 95% confidence intervals and *P*-values were determined by Kruskal–Wallis tests with Dunn's multiple comparisons (**c**–**e**) or two-sided Mann-Whitney *U*-tests (**f**–**h**). Each data point corresponds to a single donor. Trans transitional, Mat mature, cMBC classical memory B cells, atBC atypical B cells, ASC antibody-secreting cells.

---

indicative of an active, extrafollicular B cell response during active INS. We confirmed the expansion of atBCs by flow cytometry (T-bet⁺ CD11c⁺ CXCR5⁻ CD27^{+/-} CD21^low CD20⁺ CD38^{-/low}) in 13 children with active INS. A recent report characterizing circulating B cells in patients with GC-sensitive INS by cytometry by time-of-flight (CyTOF) also showed an expansion of a CD11c⁺ T-bet⁺ B cell population, though this population was not discussed by the authors[62]. Moreover, as both MZ-like B cell and atBC populations harbor autoreactive clones[61], the expansion of these populations highlights the potential for autoreactivity through the extrafollicular pathway in pediatric INS. Of note, trajectory inference predicted that MZ-like B cells may act as precursors for atBCs, consistent with a recent report in malaria-infected adults[27]. To investigate this relationship between MZ-like B cells and atBCs, future work should decipher individual B cell clonotypes operating in active INS.

Extrafollicular responses do not undergo the same degree of class-switch recombination and somatic hypermutation as germinal center-dependent follicular responses resulting in the rapid generation of IgM-secreting short-lived ASCs[22]. T-bet, a transcription factor that promotes conversion into ASCs[63], was more strongly expressed in INS atBCs than atBCs from healthy children, thus further supporting their participation in antibody-producing responses. Accordingly, we also observed a strong expansion of actively cycling T-bet⁺ ASCs in INS. Unlike ASCs in HC, INS-associated ASCs were skewed towards an

unswitched phenotype (IgM⁺ IgD⁺) consistent with an extrafollicular origin. Both atBCs and ASCs also showed lower expression of FcRL5, a receptor with ITIM domains that restricts BCR signaling[59], in INS children than HC suggesting a possible mechanism for the dysregulated activation of atBCs and their differentiation into ASCs. Our findings support earlier reports which demonstrated ASC expansion in adult and childhood GC-sensitive INS[13,14], and further point to the extrafollicular reaction as a possible mechanism for the generation of podocyte-targeting autoantibodies that have been recently reported in subsets of children with INS[15–17].

Evidence for an extrafollicular response was also found within the INS-associated gene signatures of B cell subclusters. The marked upregulation of genes associated with atBCs and the preferential expression of *GPR183* (EBI2) and *TNFRSF13B* (TACI) in naïve B cells and recently activated B cells shows that this preference for the extrafollicular response takes place early during B cell activation. Indeed, *TNFRSF13B* was specifically upregulated in INS B cells over HC B cells and was amongst the most highly enriched genes in the nephrotic B cell signature. Cell network analysis using CellChatDB showed that the APRIL signaling pathway was markedly elevated in INS PBMC. In INS, monocytes were identified as a major putative source of APRIL signal in INS, consistent with a recent report that monocyte-derived dendritic cells provide APRIL to promote the extrafollicular generation of ASCs[49].

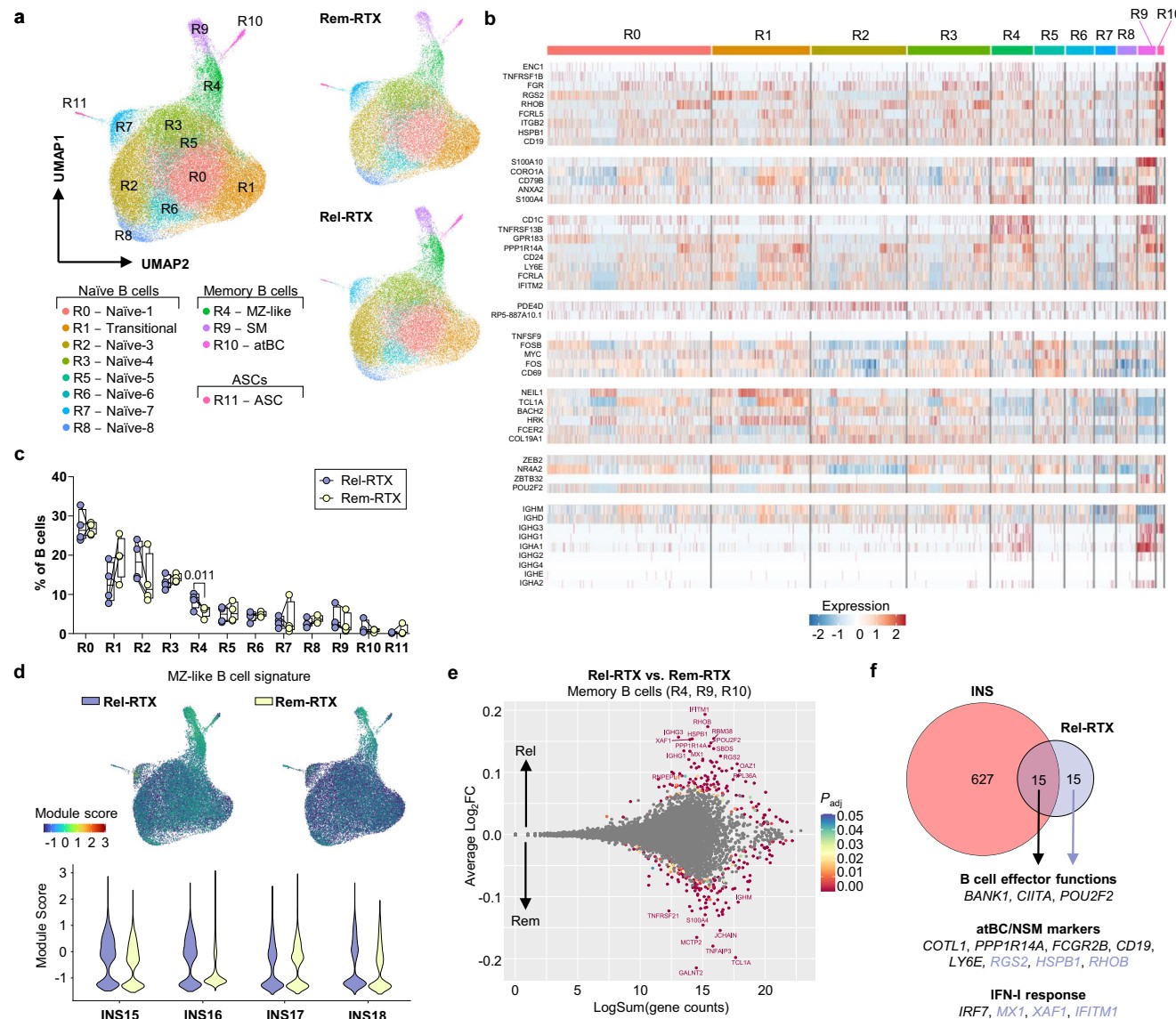

**Fig. 6 | Post-RTX relapses are associated with a nascent resurgence in extra-follicular B cells. a** Integrated UMAP of total B cells (*left*) obtained during B cell recovery from four individuals who relapsed following RTX treatment (Rel-RTX, $n = 4$) and were subsequently treated with GC and RTX to maintain long-term remission (Rem-RTX, $n = 4$). Rel-RTX, relapse following rituximab; Rem-RTX, remission maintained by rituximab **b** Heatmap showing expression of B cell sub-cluster genes identified in Fig. 2c. **c** Proportions of B cell clusters. Data are shown as box plots with median (center), interquartile range (bounds of box), and min-max range (whiskers); *P*-values were determined using individual paired two-sided *t*-tests. Each pair of dots corresponds to a single donor sampled at Rel-RTX and Rem-RTX time points ($n = 4$/group). **d** Feature and violin plots showing the module score

for the MZ-like B cell signature in total B cells. **e** Differential gene expression ana-lysis between Rel-RTX and Rem-RTX memory B cells was performed using the Seurat FindMarkers function. MA plot showing the differential expression of genes between Rel-RTX and Rem-RTX memory B cells. **f** A Venn diagram showing the overlap between the nephrotic B cell signature and the most enriched genes in Rel-RTX B cells. Genes with |log₂(Fold Change)| > 0.1, $P_{adj}$ < 0.01 and a minimum percent expression of 15% were considered differentially expressed; $P_{adj}$ values were determined using Wilcoxon Rank Sum tests with Bonferroni correction. MZ-like, marginal zone-like B cells; SM, isotype-switched memory B cells; atBC, atypical B cells; ASC, antibody-secreting cells.

The first episode and relapses of INS are usually preceded by a triggering immune event, often involving respiratory viral infections caused by influenza, parainfluenza, respiratory syncytial virus, adeno-viruses, and, most recently, SARS-CoV-2[34–36,64]. Although the mechanism by which viral infections trigger relapse is unknown, viral infections are effective activators of humoral immunity in an IFN-γ- and type-I IFN-dependent manner[65]. Using scRNA-seq, we showed a possible role for type-I IFN signaling in driving the nephrotic B cell response. Interest-ingly, 5/13 children studied with active INS experienced upper respira-tory tract infections in the 2 weeks prior to proteinuria onset (Supplementary Data 1). While we cannot rule out that the expansion of atBCs and ASCs defined herein may be driven by viral infections, future

work will aim to determine whether these infections may precipitate INS relapses through the stochastic activation of autoreactive B cells.

Finally, we sought to investigate the impact of immunosuppres-sive treatment on the INS-associated B cell populations that we uncovered. While GC treatment specifically restricted the abundance of ASCs in the blood, RTX effectively ablated all INS-associated B cell populations and maintained B cells in a transitional naïve state well into the recovery phase. The broader and more sustained effects of RTX on B cell compartments may explain the longer remission times observed with GC/RTX combination therapy than GC alone[6]. Addi-tionally, the ability of RTX to ablate ASCs in INS supports that these cells are short-lived ASCs that developed extrafollicularly.

As with RTX-maintained remission, post-RTX relapses also featured a highly naïve B cell profile. We therefore hypothesized that these relapses were taking place because of a nascent version of the nephrotic B cell response. We confirmed earlier findings that the post-RTX relapses were associated with a resurgence in isotype-switched cMBCs but were unable to detect an expansion of atBCs or ASCs in this setting[20]. A possible explanation for this observation is that post-RTX relapses may represent an early time point in the extrafollicular response before the export of pathogenic B cells to the peripheral blood. Supporting this, scRNA-seq of four children from whom we were able to acquire post-RTX relapse-remission paired PBMC revealed that there was an expansion of MZ-like B cells in post-RTX relapse, the memory B cell population predicted to act as an atBC precursor by trajectory inference. This finding was confirmed by flow cytometry with an expansion of IgM$^+$ IgD$^+$ cMBCs, a heterogeneous population that contains MZ-like B cells, in post-RTX relapse samples[46]. Additionally, post-RTX relapses may be associated with the accumulation of clonally expanded autoreactive B cell populations that persist following RTX treatment, the detection of which would warrant clonotyping of post-RTX B cells. Indeed, the post-RTX persistence of autoreactive memory B cells in the blood has been reported in SLE and ANCA-associated vasculitis[66], in the spleen in immune thrombocytopenia[67], and in the lymph nodes in kidney transplant recipients[68].

Differential gene expression analysis between post-RTX relapse and remission did not identify any significant differences when completed at the pseudobulk level. We reasoned that this was due to the paucity of pathogenic B cells in the naïve-enriched post-RTX setting, a scarcity that was only compounded by the limited number of cells that we could analyze by scRNA-seq. To this end, we also conducted differential gene expression analysis by comparing all post-RTX relapse memory B cells to post-RTX remission B cells, without pseudobulk separation. Here, we identified significant enrichment of nephrotic B cell signature genes and genes relating to type-I IFN signaling in memory B cells during post-RTX relapse. However, as these differences were not consistently observed across donors, the post-RTX transcriptional signature needs to be verified with memory pre-enriched B cells derived from a greater number of children.

In summary, our results uncovered a previously unrecognized role for extrafollicular B cells in childhood INS. We propose that these B cells contain autoreactive clones that may give rise to ASCs that produce podocytopathic autoantibodies thereby contributing to the pathogenesis of INS. This work provides a rationale for further exploration of B cell-targeting therapeutics in pediatric INS, in particular the targeting of atBCs alongside other autoimmune conditions.

## Methods

### Human participants and PBMC collection

This observational longitudinal study follows thirty-one children with INS that were enrolled according to protocols approved by institutional Research Ethics Boards (REBs) at the Research Institute of the McGill University Health Centre (MUHC-14-466, T.T.) and the Alberta Children's Hospital (CHREB-16-2186, S.S.)[69]. Briefly, parents/legal guardians were informed of the study and provided their written consent for blood collection and the use of the sample in our research by signing an informed consent form. Additionally, children between the ages of 7 and 18 years were informed of the study and were asked to sign an assent form. For children with INS, samples were collected either during hospital visits necessitated due to a proteinuric relapse or a scheduled visit for follow-up or for treatment while in remission. Blood samples were collected during active disease (first onset or relapse; defined by a urinary protein-to-creatinine ratio (uPCR) ≥ 0.2 g/mmol, serum albumin ≤25 g/L, and edema), and in remission (negative/trace dipstick or uPCR ≤0.02 g/mmol). Samples from children with active INS were stratified into RTX-inexperienced

(INS, $N = 14$; median age of 8.1 years, interquartile range 5.5–10.8 years; 7 females) and -experienced (Rel-RTX, $N = 7$; 9.7 years, IQR of 9.0–11.0 years; 2 females) groups based on previous exposure to RTX. In the RTX-inexperienced group, 12/13 children had GC-sensitive INS (1/12 biopsy proven FSGS) and one child was diagnosed with GC-resistant membranous nephropathy (identified by the yellow point in relevant graphs). Of these, 8/13 children were untreated at the time of sampling, the remainder received weaning doses of prednisone. In the RTX-experienced group, all children had GC-sensitive INS (2/6 biopsy proven MCD, 1/6 FSGS). Five of the RTX-experienced GC-sensitive INS samples were obtained while off therapy and the remaining sample was taken during a prednisone taper. Samples from children in remission were stratified into GC-induced remission (Rem-GC, $N = 13$; 9.9 years, IQR of 6.2–12.5 years; 6 females) and remission maintained by RTX (Rem-RTX, $N = 14$; 9.2 years, IQR of 7.0–11.1 years; 5 females). Rem-GC samples were taken after completion of prednisone taper, though two patients were receiving mycophenolate mofetil at the time of sample collection, and Rem-RTX samples were collected following the re-emergence of CD19$^+$ cells off therapy. For healthy controls, we enrolled twenty-four healthy children undergoing minor day surgery or healthy volunteers from the community into the study (11.6 years, IQR of 8.4–14.6 years; 13 females). All samples were used in accordance with our standard operating protocol (MUHC-15-341, T.T.).

### Single-cell and RNA preparation from PBMC

Cryopreserved PBMC were thawed and rested for 2 h at 37 °C and in 5% CO$_2$. Cells were then washed and resuspended in PBS + 2% FBS before fluorescence activated cell sorting using the BD FACSAria Fusion. Live cells from PBMC of HC ($N = 4$) and INS ($N = 4$) children (HC-INS scRNA-seq dataset) were isolated based on size and granularity, while live B cells (CD19$^+$ CD4$^-$ CD8$^-$) were sorted from Rel-RTX ($N = 4$) and Rem-RTX ($N = 4$) PBMC (Rel-Rem scRNA-seq dataset). Sorted cells were washed twice in PBS + 0.04% BSA and were brought to a concentration of 1000 cells/µl. Samples were subsequently processed according to the 10x Genomics Single Cell 5' v1.1 user guide. Single-cell PBMC suspensions were loaded onto the 10x Single Cell Chip G along with 10x Genomics NextGem scRNA 5' V1.1 reagent. We targeted 10,000 captured events on a 10x Genomics Chromium controller. Complementary DNA (cDNA) and 5' gene expression libraries were generated using the standard 10x Genomics protocol. Libraries were sequenced on a NovaSeq6000 (Illumina).

### scRNA-seq data preprocessing and quality control

Raw FASTQ files were aligned to the GRCh38 reference genome and count matrices for cell barcodes and UMIs were generated for each sample by CellRanger (v3.0.1). Using the Seurat (v4.3.0) R package, samples were filtered from cells expressing any two lineage markers (CD79A, CD3G, CD14, and LILRA4) as these were considered doublets[70,71]. In the HC-INS dataset, doublet and nonviable cells were removed by excluding cells expressing <200 or >3000 genes, >10% mitochondrial genes, and <10% ribosomal protein genes. In the Rel-Rem dataset, cells expressing <200 or >2500 genes, >10% mitochondrial genes, and <7% ribosomal protein genes were filtered. Lingering doublets were predicted and removed using the DoubletFinder (v2.0) R package[72].

### Generating scRNA-seq clusters

To generate clusters that were uniformly present in all samples, we used the reciprocal PCA method for integrated clustering in Seurat. Data were normalized and variable features were identified for each sample separately. Integration features were identified (SelectIntegrationFeatures), scaled, and PCA was performed. Integration anchors were then generated from the integration features (FindIntegrationAnchors) and two integrated Seurat objects for the HC-INS and Rel-Rem datasets were produced (IntegrateData). The integrated

objects were subsequently scaled on variable features and PCA was performed. UMAP clustering was completed using 35 (HC-INS) or 10 (Rel-Rem) PCs at a resolution of 0.5. For the Rel-Rem dataset, a single cluster comprised of contaminating T cells (expressing *CD3G* or *CD3E*) were removed, and the object was then re-clustered using 10 PCs and a resolution of 0.5.

For the HC-INS object, cluster identities were determined by expression of major lineage defining genes (Supplementary Fig. 1b). To generate B cell subclusters, the Bnaive, Bmem-1, Bmem-2, and ASC clusters were isolated, and re-clustered using the reciprocal PCA method with 10 PCs and a resolution of 0.5. Two small contaminating T and NK cell clusters (expressing *CD3G*, *CD3E*, or *NKG7*) were removed, and clustering was performed again.

### Differential gene expression
Pseudobulk differential gene expression analysis was completed using the muscat R package (v1.14)[37]. First, the HC-INS Seurat object was converted into a Single Cell Experiment (SCE) object and gene counts were normalized and log-transformed. Counts were then aggregated at the level of the broad immune cell lineages (CD4$^+$ T cells, CD8$^+$ T cells, NK cells, NKT cells, B cells, and monocytes) for each sample using the aggregateData function in muscat and differential gene expression analysis between INS and HC individuals was carried out using edgeR. Gene lists for each broad immune cell lineage was filtered from genes present in <10% of cells and genes with a |log$_2$FC| < 0.65 (Supplementary Data 2). The genes upregulated in INS B cells comprised the nephrotic B cell signature while downregulated genes comprised the healthy B cell signature. To determine B cell subcluster identities, differential gene expression analysis between individual subclusters was done using the FindMarkers function in Seurat v4. Pseudobulk differential gene expression analysis using the muscat R package (v1.14) was used as outlined above to identify differentially expressed genes between INS and HC in each B cell subcluster. For the Rel-Rem Seurat object, differential gene expression analysis was performed between Rel-RTX and Rem-RTX memory B cells using the FindMarkers function in Seurat v4 (Supplementary Data 6).

### Pathway analysis, gene set enrichment analysis, and module scores
Pathway analysis on the nephrotic B cell signature and the INS-associated signature for each B cell subcluster was performed using g:Profiler with the Gene Ontology Biological Process and Reactome databases[73]. Terms that were significantly enriched ($P_{adj}$ < 0.05) and consisting of at least five and no more than 500 genes were organized in 2D space using the EnrichmentMap plugin on Cytoscape v3.9.1[74]. Enriched terms were clustered into groups using the AutoAnnotate and ClusterMaker2 plugins. GSEA of Gene Ontology terms in unfiltered gene lists organized by log$_2$FC was carried out using the GSEA App (Broad Institute).

### Trajectory inference analysis
The ASC subclusters (B8 and B9) were removed and the remaining B cell subclusters (B0-B7) from both HC and INS individuals were re-clustered. Pseudotemporal trajectories were constructed on the new UMAP using the Monocle3 R package[48]. Pseudotime was calculated by selecting all trajectory nodes within the transitional naïve B cell subcluster as the starting point. Cells along distinct branches were isolated for downstream comparisons of *TNFRSF13B* and *GPR183* expression.

### Cell network analysis
CellChatDB (v1.6) was used to infer cell networks between clusters in INS and HC PBMC following the accompanying tutorial[51]. Briefly, CellChatDB objects were generated from Seurat objects and over-expressed genes in each cluster were identified. The probability of each cluster in participating as a sender, modulator, or receiver of each signal was determined using known networks of ligands, receptors, and co-factors. INS and HC PBMC were analyzed independently. We show the results for the APRIL signaling pathway.

### Flow cytometry
Cryopreserved PBMC were thawed, rested, washed twice with PBS + 2% FBS, and counted using the hemocytometer. Over 90% viability was confirmed by trypan blue and no more than $1 \times 10^6$ cells were stained per flow cytometry panel. Cells were then incubated with Fixable Viability eFluor 780 Dye (ThermoFisher Scientific) at 4 °C for 15 min, washed with PBS + 2% FBS, and incubated for another 15 min at 4 °C in the presence of Fc receptor block (BD Biosciences). Antibody cocktails for surface proteins were prepared in PBS + 2% FBS and Brilliant Stain Buffer (50 μl/100 μl of cocktail, BD Biosciences) and added to the cells. Cells were incubated at 4 °C for 20 min before washing with PBS + 2% FBS and fixation/permeabilization was performed using the eBioscience Foxp3/Transcription Factor Staining Buffer Set (eBioscience). Cells were then washed with 1X permeabilization buffer (eBioscience) and incubated for 45 min at 4 °C with antibody cocktails detecting cytoplasmic and nuclear proteins. Two final washes, the first in 1X permeabilization buffer and the final in PBS + 2% FBS, were performed before cells were acquired on the BD LSRFortessa X-20.

Extracellular staining was performed using the following antibodies: anti-human CD3ε BV785 (1:50, OKT3, BioLegend), anti-human CD19 BV605 (1:20, SJ25C1, BD Biosciences), anti-human CD20 Alexa Fluor 700 (1:50, 2H7, BioLegend), anti-human CD21 BV421 (1:20, B-ly4, BD Biosciences), anti-human CD27 PE-Cy7 (1:20, M-T271, BD Biosciences), anti-human CD10 BUV737 (1:20, HI10a, BD Biosciences), anti-human CD38 BUV737 (1:20, HB7, BD Biosciences), anti-human CD11c PerCp-Cy5.5 (1:20, B-ly6, BD Biosciences), anti-human IgD BV510 (1:20, IA6-2, BioLegend), anti-human IgM Alexa Fluor 488 (1:40, MHM-488, BioLegend), anti-human FcRL5 APC (1:20, 509f6, BioLegend), anti-human CD24 PE (1:20, ML5, BD Biosciences), anti-human CD1c BV711 (1:20, L161, BioLegend), and anti-human CXCR5 APC (1:20, J252D4, BioLegend). Intracellular proteins were stained using the following antibodies: anti-human T-bet PE (1:20, 4B10, BioLegend), and anti-human Ki-67 BUV395 (1:50, B56, BD Biosciences). Data were analyzed on FlowJo v10.8 software (FlowJo, LLC).

### FlowSOM clustering
Unsupervised high-dimensional clustering of B cells was done using the FlowSOM plugin for FlowJo v10.8 software[75]. We randomly selected 15,000 or 8125 B cells (live CD19$^+$ CD3$^-$) from 13 RTX-inexperienced children with active INS and 24 HC, respectively, for concatenation (195,000 INS B cells, 195,000 HC B cells). FlowSOM clustering was performed using ten B cell surface markers: CD19, CD20, CD21, IgD, IgM, CD27, CD38, CXCR5, CD1c, and CD24. Briefly, each B cell was plotted onto a 10 × 10 self-organizing map (SOM) yielding 100 distinct clusters. Minimal-spanning trees (MST) were built upon these clusters with branches containing cells with similar surface phenotypes. The clusters were further categorized into 14 metaclusters, the identities of which were determined by the expression of the ten surface markers.

### Statistical analysis
Statistical analyses were performed on R or using the GraphPad Prism v9 software. Two-sided analyses between two groups were done using Mann−Whitney *U*-tests. For any comparison between more than two groups, a Kruskal−Wallis test was employed with multiple comparison's (Dunn's test). A two-way ANOVA was used for comparisons of two parameters between two groups with multiple comparisons (Tukey's test). Finally, a paired two-sided *t*-test was used for longitudinal analyses. Data are shown as median with 95% confidence intervals with each point representing a single study participant or box plots with the center at the median, the edges of the box at the interquartile range, and the whiskers at data minima and maxima. For

scRNA-seq data, $P_{adj}$ values were determined using the edgeR method with Benjamini-Hochberg correction for pseudobulk differential gene expression analyses or Wilcoxon Rank Sum tests with Bonferroni correction for non-pseudobulk differential gene expression analyses (e.g., to obtain markers for cell clusters). $P$-values < 0.05 were considered significant and significant $P$-values were depicted on graphs. No sample size calculations were conducted a priori. All samples obtained were used for the study.

### Reporting summary

Further information on research design is available in the Nature Portfolio Reporting Summary linked to this article.

## Data availability

Raw sequencing data and the processed count matrices included in this study were deposited in the NCBI Gene Expression Omnibus (GEO) as a super series under the accession number GSE233277. The results of all differential gene expression analyses, pathway analysis, and transcription factor enrichment analysis are included in Supplementary Data files of this paper. All source data needed to evaluate the conclusions are included with this paper. Public data repositories used for transcription factor enrichment and pathway analysis include ChEA3[41] and g:Profiler[73] (https://biit.cs.ut.ee/gprofiler/gost), respectively. The GRCh38 human reference genome was used for sequence alignment. Source data are provided with this paper.

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

## Acknowledgements

We are grateful to all study participants, their families, and the medical teams involved in their care. We thank the Canadian Childhood Nephrotic Syndrome Study (CHILDNEPH) for supplying PBMC, D. Muruve and the Biobank for the Molecular Classification of Kidney Disease

(BMCKD) for processing and storing PBMC, and B. Mazer and B. Foster for supplying healthy children PBMC. We also thank I. Ragoussis, H. Djambazian, and Y. Wang at the McGill Genome Centre for sample processing through the 10X Genomics platform, sequencing, and generation of pre-processed count matrices, and M-H. Lacombe, E. Iourtchenko, and H. Pagé-Veillette at the Research Institute of the McGill University Health Centre Immunophenotyping Core for their support. This work was supported by a Canadian Institute of Health Research (CIHR) project grant (PJT-166006 to T.T., C.P., S.S.) and a Kidney Foundation of Canada Kidney Health Research grant (KHRG-190010 to T.T., C.P., S.S.). T.A. was supported by a CIHR Doctoral Award and a FRQS Doctoral Award.

## Author contributions

T.A., C.A.P. and T.T. conceptualized the study and designed all the experiments. T.A. conducted the experiments and performed all the data analysis. T.A., L.A. and M.A.P. processed all the blood samples. G.P. and M.A.P. obtained consent from the parents/legal guardians of all participating children and further obtained assent from subjects between 7 and 18 years of age. T.A., L.A., M.A.P., M.B., S.M.S. and T.T. coordinated sample collection and biobanking at the McGill University Health Centre and Alberta Children's Hospital. T.A., D.L., C.A.P. and T.T. planned all the bioinformatics analyses. T.A., C.A.P. and T.T. wrote the initial draft of the manuscript. All authors critically reviewed the manuscript, discussed the results, interpreted the data, and contributed to the formulation and agreed on the submission of the final draft of the manuscript.

## Competing interests

The authors declare no competing interests.
