## [Peer Review File · Nature Communications]

REVIEWER COMMENTS

Reviewer #1 (expert in nephrology):

In the present manuscript by Al-Aubodah and colleagues, the authors analyze PBMCs from 4 patients with INS and 4 matched controls using scRNA-seq. They find a dominant expansion of B cells in patients with INS, which was associated with the largest number of transcriptional changes in these cells (958 of 1976 genes). Of the 958 differentially expressed genes in B cells, 642 were upregulated, defining the “nephrotic B cell signature” in patients with INS. This signature involved BCR signaling, B cell activation, IgG production, oxidative phosphorylation, fatty acid oxidation, antigen presentation, and actin cytoskeleton organization. Induction of these pathways was licensed by upregulation of corresponding TFs. Further analyzing the subclusters in the investigated cohort, the authors found a reduction of naïve B cells and increases in memory B cells and ASCs in INS samples. The expanded clusters were further defined as MZ-like BCs and atBCs. These two clusters expressed the TF POU2F2, one of the TFs linked to the nephrotic B cell signature. Moreover, the “nephrotic gene signature” of 642 upregulated genes was most enriched in the MZ-like BCs, switched memory B cells and atBCs, demonstrating that this signature is most present in the expanded MZ-like BCs and atBCs. The authors further show that this INS B cell response involves the characteristics of an extrafollicular pathway, e.g. through monocyte/DC-derived APRIL.

The authors then continue their investigations using flow cytometry and found that most of the expanded clusters were actively cycling CD21^{low} memory B cells and ASCs. The CD21^{low} cells were CD11c⁺ and Tbet⁺, indicating that the expanded CD21^{low} cells in INS are atBCs.

Next, the authors evaluated the consequences of GC versus GC/Rtx treatment on BC populations in INS. GCs reduced ASCs but not other BC subpopulations, while Rtx reduced BCs such as cMBCs and atBCs. Interestingly, patients who relapsed after Rtx therapy still had reduced numbers of total B cells, but increased numbers of non-switched and switched cMBCs (while atBCs and ASCs were unchanged in Rtx-induced remission and relapse after Rtx).

Finally, the authors perform scRNA-seq in 4 patients after Rtx-induced remission and subsequent relapse and find that MZ-like BCs/atBCs, expressing genes of the nephrotic signature, were expanded during relapse after Rtx therapy.

Even though the study completely lacks functional data and is largely descriptive, I think that the presented analyses describing the expansion of MZ-like and atBCs in INS patients provide many novel insights into the pathogenesis of INS. I particularly appreciate the analyses involving GC versus Rtx treatments and relapse after Rtx therapy. Overall, I think that this is an excellent study.

I have some comments that may help to further improve the manuscript.

Specific comments:

1. In Fig. 1b and Supplementary Fig. 1, the authors show that B cell lineage is the only lineage which is preferentially expanded in children with INS. Based on the cell proportion in Fig. 1b, it looks like NK cells similarly increased (yet no p value is presented in Fig. 1b). How do the authors explain this? Is this a technical issue or a true biological effect, which can be further explored? Can the authors show the proportions of subclusters in B cells and other main cell clusters (like CD4+ T, CD8+ T, NK etc.) between HC and INS? Are the proportion changes in B cell clusters in Fig. 1 corresponding with the proportion changes of B cell subclusters in Fig. 2b?

2. Regarding the B cell subclusters in Fig. 2, is it possible to map the datasets to other publicly available scRNA-seq datasets from healthy PBMCs, or PBMCs from patients with other autoimmune diseases, in order to provide further evidence of for atypical B cells and MZ-like B cells?

3. In Fig. 2e, what are the details of the nephrotic and healthy B cell signatures? What kind of reference genes are used to score the nephrotic signatures of B cell subclusters?

4. In Fig. 6e, the authors performed the comparison of differential gene expression of all memory B cells (clusters R4, R9, and R10) between relapse and remission timepoints. How about the differentially expressed genes of one single memory B cell cluster between relapse and remission groups?

5. I am a little lost with the analyses in patients with a relapse after Rtx therapy. I understand that there is a difference between the flow cytometry analysis and the scRNA-seq (atBCs were unchanged in flow cytometry but MZ-like BCs, the predicted precursors of atBCs, were expanded in scRNA-seq analyses). How do the authors explain this? Is it because MZ-like cells cannot be analyzed using cytometry? Or is the scRNA-seq analysis is more sensitive? Or where these different patients? These circumstances are not entirely clear to me. Please better explain, also in the discussion.

6. What could be the possible reasons for the relapse of INS patients after Rtx treatment, since the ASCs and atBCs are not significantly expanded in relapsed patients? In addition, did the naïve B cell clusters show some differences in gene expression profiles between the relapsed and remission samples?

7. Some of the legends miss details on the statistical tests, sample numbers etc.

Reviewer #2 (expert in B cells in nephrotic syndrome):

I read with interest the manuscript entitled “The extrafollicular B cell response is a hallmark of childhood idiopathic nephrotic syndrome”. In this study, authors aimed at characterizing by scRNAseq analysis the immune transcriptional signature in INS pediatric patients, both in active phase and in remission before or following glucocorticoid therapy or rituximab, compared with age-matched healthy controls. Authors observed that, among the major immune cell populations, B cells are the only lineage expanded in INS patients. By evaluating the B-cell specific transcriptional signatures, authors found that B cells from INS patients have an elevated expression of genes involved in activation, and differentiation, BCR signaling, antibody production, oxidative phosphorylation, actin polymerization and type I IFN signature. They also found an enrichment in memory but not in naïve B cell subsets, mainly in marginal zone-like (MZ-like), somatic-mutated (SM) memory and atypical memory B cells as well as in antibody-secreting B cell (ASC) subsets for INS patients. These results were validated by flow cytometry analysis and additional info on the effect of glucocorticoid (GC) or rituximab (RTX) treatment on the amount of the different B-cell subtypes were provided, showing as GC treatment only affects ASC levels in contrast to RTX treatment, which affects the entire B-cell repertoire. Finally, a comparison between patients who relapsed or were in remission following RTX treatment showed as the re-emergence of memory B cells, in particular of the switched isotype, was associated with relapse, with an enrichment also in both MZ-like and atypical B cells.

The manuscript is very well written and results are clearly described and discussed. Together with a validation of previous findings, novel interesting data were reported, with a strong contribution to the increase of the knowledge of the pathogenic mechanisms underlying INS, which are still not well elucidated.

I have only some suggestions to improve the manuscript:

- the selection of the patients is very rigorous and a detailed description of the characteristics of each patient is reported in supplementary Data 1; however, this description cannot be easily extrapolated and details on the state of the disease as well as on the current medication at time of sampling is not so clear by reading the text. I would suggest to add a Table resuming patient characteristics sub-grouped as described in figure 5a (INS, Rem-GC, Rel-RTX, Rem-RTX) by reporting the most relevant info (mean-SD age, # of patients at onset or in relapse for patients in active phase, # of patients on current or previous medication, mean time from last medication when administered (mainly for GC and RTX), mean-SD uPCR, co-morbidity, RNAseq or flow cytometry analysis) for each subgroup.

- what are the inclusion criteria for healthy donors? Where are they enrolled? Are they patients followed at Institution for other non-immunological conditions? Please specify.

- For patients treated with RTX, relapsing patients have a re-emergence of isotype-switched memory B cells more than atypical memory B cells or ASCs. In addition to the hypothesis reported in the current manuscript, this association could be due to the persistence of isotype-switched and clonally expanded memory B cell clones, as previously reported in other autoimmune diseases (Bashford-Rogers RJM, et al. Nature. 2019. doi: 10.1038/s41586-019-1595-3). This could also be due to the persistence of isotype-switched memory B cell clones in secondary lymphoid organs, which are not effectively depleted by

rituximab treatment (Wallin et al. Blood. 2014. doi: 10.1182/blood-2014-07-585976). Both papers should be commented and reported in the discussion. In this regard, the characterization of the BCR repertoire as performed by Bashford-Rogers et al. in longitudinal samples could be very helpful to verify also in the setting of INS whether the post-RTX re-emerging B cell subsets are persistent or newly emerging clones.

Minor issues:

- the amount of total B cells, as determined by flow cytometry, should be reported by comparing INS patients and HC, to verify the results obtained by RNAseq.

- in supplemental figure 4a is reported the intensity of CD19 expression on the surface of CD21^{low} B cells. Why the expression of CD19 is significantly lower in INS patients compared to HC? Please describe and discuss it.

Reviewer #3 (expert in B cells in autoimmune disease):

This is an excellent study.

I do have some issues that the authors might want to consider. Fig 1 shows an INS signature. It probably should be noted how much of this is a consequence of a different frequency of B cell subsets (shown in Fig 2) rather than DEGs within subsets.

The study of those who relapse after rituximab is of great interest. The authors discuss the difference in frequency of MZ B cells. They might want to focus on the difference in frequency of transitional and naïve B cells as this probably reflects BAFF levels and belimumab might be of therapeutic use.

Reviewer #4 (expert in single-cell RNA sequencing):

Please see attachment.

Review of the manuscript entitled “**The extrafollicular B cell response is a hallmark of childhood idiopathic nephrotic syndrome.**” by Al-Aubodah for Nature Communications. In this study, the authors performed an analysis on two independent scRNA-seq datasets, one from PBMCs including pediatric INS donors and healthy controls and the other from B cells with different treatment responses. They described the transcriptome difference in B cells between INS and control. The expansion of atypical B cells (atBCs), marginal zone-like B cells, and antibody-secreting cells (ASCs) were identified in INS, indicating the engagement of an extrafollicular B cell response. The authors also performed flow cytometric analysis and demonstrated that Proliferating T-bet+ atBCs and ASCs are a hallmark of active INS. The study indicated the clinical role for extrafollicular B cells as potential origin for autoreactive ASCs in childhood INS.

I should say upfront that I am not a B cell expert and also not particularly versed in INS (MCD), so it is hard for me to judge the biomedical novelty of this study. The main novelty of this study, in my eyes, seems to be that the authors found extra-follicular B cells expanded, whereas prior studies found memory B cells expanded which suggests a classical follicular B cell response with ASC generation. No investigation into the role of these cells has been conducted, so the article is a description of the B cell changes in blood in INS, and the link to B cell caused pathology due to effective rituximab treatment has been put into question recently (<https://www.nature.com/articles/ncprheum0424>). As a more technical person, I will therefore focus my attention on the technical side of this manuscript.

Major points

1. Absence of single cell quality controls: There is no information given to prove the donor/sample quality are comparable, such as the number of genes/UMIs detected and percentage of mitochondrial genes. This information is important for any downstream analysis and comparison. For example, in Fig. 1B the author compared the major cell lineage proportions between INS and HC and concluded the major proportion difference is B cells, while it showed one single sample is dominating the comparison for B cells.
2. Missing information on unbiased cluster makers: For both of the two clustering annotation analyses (PBMC and B cells), the authors didn't provide the top marker genes lists for all the clusters. For all PBMC clusters, the authors only showed selected canonical lineage markers in

Fig. S1B but did not optimize the order of the markers. For the subclusters of B cells, the marker genes were missing. The authors showed the heatmap for memory cell clusters (Fig. 2D), but not for other B cell clusters (B 0,4,7,8,9). Please include the complete top makers of all clusters in Fig. S3.

3. For the pseudobulk differential gene expression analysis, the authors should also set a p-value threshold to determine the significantly changed genes instead of only using log2FC and percentage of expression.
4. Fig. S1C showed that only 381 differentially expressed genes were unique for B cells, the remaining 577 genes were shared between B cells and other cell types. For the pathway analysis, the authors should explore which pathways are unique for B cells and which ones are shared between B cells and other immune cells. This is the advantage of this nice single cell dataset and should not be ignored.
5. The analysis in Fig. 2E is not convincing. The nephrotic signatures were calculated on pseudobulk B cells. And the bigger clusters within the B cells would contribute more to the global B cell nephrotic signatures. It seems that B3 and B5 have more cells than B6. So it is not entirely fair to say that the nephrotic signature was most pronounced in MZ-like B cells (B3) and SM B cells (B5). One alternative approach is to perform differential expression analysis between healthy and disease on each subcluster of B cells and then the enrichment of pathways for each subcluster.
6. The authors compared the features of CD21^{low} B cells identified in flow cytometry data to that of atBCs recovered from scRNA-seq data. It's not clear what the conclusion is. A better description and visualization should be provided.
7. The clustering resolutions used for different scRNA-seq analysis were all set to be 0.5. However, the separation was not optimized especially for Fig. 6. The authors should consider trying different resolutions and optimize the clustering and the annotation.
8. It would be great if the authors could validate some of the scRNA-seq results using e.g. qPCR.

Minor points

1. The two UMAPs in Fig. S1A are not sufficient to demonstrate that the 18 distinct immune cell populations were uniformly present in all donors. This can only validate that all the populations

are present at both disease and healthy conditions. Please provide a barplot of the cluster distribution across all the donors.

2. Cluster C1 in Fig. 1A and Fig. S1B showed a clear expression of Treg signatures (FOXP3, CTLA4). The current annotation of T-mem1 is less accurate.
3. Fig. S1C only shows the upregulated intersection of genes, but the figure legend doesn't mention that.
4. It would be great if the authors could reproduce the transcription factor enrichment analysis using SCENIC and provide the results in a supplementary figure.
5. In Fig. 2G the three trajectory branches are not clear.
6. The authors used "CellChat" for APRIL signaling network analysis in Fig. 2H but the corresponding methods section is missing.
7. In Fig. 3A the FlowSOM clustering result is difficult to read (especially the text coloring is hard to take in).

Response to reviewer comments

The individual comments made by the Reviewers are addressed point-by-point below:

Reviewer #1 (expert in nephrology):

1. Comment: “*In Fig. 1b and Supplementary Fig. 1, the authors show that B cell lineage is the only lineage which is preferentially expanded in children with INS. Based on the cell proportion in Fig. 1b, it looks like NK cells similarly increased (yet no p value is presented in Fig. 1b). How do the authors explain this? Is this a technical issue or a true biological effect, which can be further explored?*”

Response: We appreciate this astute observation from Reviewer 1. The proportions of total NK cells – composed of *FCGR3A* high-expressing (cluster C8, CD16^{high}) and low-expressing (cluster C9, CD16^{dim}) – were not significantly different between HC and children with INS ($p = 0.20$). However, the *FCGR3A* low-expressing NK cells (cluster C9) was the only non-B cell immune population to be expanded in INS (**Supplementary Fig. 2b, page 6**). These NK cells correspond with CD56^{bright} CD16^{dim} NK cells, a sub-population of NK cells with low cytotoxic capacity but with the potential to secrete copious amounts of cytokines. The C9 NK cells expressed genes encoding for chemokines (e.g., *XCL1*, *XCL*) and cytokines (e.g., *GZMK*, *CSF2*), while having lower expression of genes associated with potent cytotoxicity (*EOMES*, *PRF1*, *GZMB*) in comparison to C8 NK cells (**Supplementary Fig. 2c**)¹, confirming findings from a previous study in childhood INS that carried out flow-based immunophenotyping of PBMC². The cause of this CD56^{bright} CD16^{dim} NK cell expansion remains unknown, and future work will explore this question. Since CD56^{bright} CD16^{dim} NK cells are primarily located in secondary lymphoid organs (~75% in lymph nodes; ~10% in circulation)³, their expansion in the peripheral blood of children with INS may denote increased recirculation of these NK cells during active INS and, thus, a greater capacity to contribute to renal pathology or support cellular immune responses.

2. Comment: “*Can the authors show the proportions of subclusters in B cells and other main cell clusters (like CD4+ T, CD8+ T, NK etc.) between HC and INS? Are the proportion changes in B cell clusters in Fig. 1 corresponding with the proportion changes of B cell subclusters in Fig. 2b?*”

Response: We now provide the subcluster proportions of each PBMC population in **Supplementary Fig. 2b**. As is evident from this analysis, only the antigen-experienced (memory and antibody-secreting) B cell populations are expanded in INS while naïve B cells are similar in proportions to those in HCs (**Supplementary Fig. 2b**). Thus, the expansion of total B cells shown in Fig. 1b do indeed correspond with the changes outlined in Fig. 2b.

3. Comment: “*Regarding the B cell subclusters in Fig. 2, is it possible to map the datasets to other publicly available scRNA-seq datasets from healthy PBMCs, or PBMCs from patients with other autoimmune diseases, in order to provide further evidence of atypical B cells and MZ-like B cells?*”

Response: The atBC signature is well established in the literature, and **we now provide GSEA plots of the atBC signatures reported in Portugal *et al.*⁴, Sutton *et al.*⁵, and Holla *et al.*⁶**; these show a clear enrichment of atBC genes within our B cell cluster B6 (**Supplementary Fig. 4c**). For cluster B3, which we termed MZ-like B cells, cells expressed a cMBC-like transcriptional profile while being predominantly isotype non-switched (expressing *IGHM*) and expressing genes associated with MZ B cells including *CD1C*, *CD24*, *PLD4*, and *MZB1*. To our knowledge, the only group to identify a similar cell is Holla *et al.*, which they termed MBC-like⁶. In accordance with our MZ-like B cells, the MBC-like cells described by Holla *et al.* shared a core MBC signature while expressing *IGHM*, *GPR183*, and *TNFRSF13B*. Moreover, using trajectory analysis, Holla *et al.* demonstrated that these MBC-like cells may act as precursors to atBC, a finding consistent with our trajectory analysis in Fig. 2f.

4. Comment: “*In Fig. 2e, what are the details of the nephrotic and healthy B cell signatures? What kind of reference genes are used to score the nephrotic signatures of B cell subclusters?*”

Response: In the submitted version of Figure 2, the nephrotic and healthy B cell signatures referred to the 642 genes upregulated in B cells in INS and the 316 genes downregulated in B cells in INS, respectively, identified by pseudobulk differential gene expression analysis (**Fig. 1d**). However, as Reviewer 4 pointed out, this analysis reflected changes in B cell proportions more so than the contribution of each B cell subcluster to the nephrotic and healthy B cell signatures. As such, we eliminated this analysis and replaced it with pseudobulk differential gene expression analysis conducted on each B cell subcluster (**Fig. 2e, Supplementary Fig. 5**). With this analysis, we now show that across all B cell subclusters, INS B cells preferentially expressed genes associated with cytoskeletal dynamics, which is likely related to the activated state of these cells (**Supplementary Fig. 5b**). More specifically, both naïve and memory B cells situated along the extrafollicular/atBC-associated trajectory (actMBCs, MBC-2, and MZ-like B cells), preferentially expressed genes associated with B cell activation and B cell receptor signaling further underlining the progression of B cells through an extrafollicular pathway during active INS (**Supplementary Fig. 5b**). Accordingly, even in the naïve state, INS B cells preferentially expressed genes promoting extrafollicular B cell reactions (*TNFRSF13B*, *GPR183*, and *POU2F2*) (**Fig. 2e**). **Figure 2 and the associated text on page 11 have been modified to address this point and Supplementary Figure 5 has been added.**

5. Comment: “*In Fig. 6e, the authors performed the comparison of differential gene expression of all memory B cells (clusters R4, R9, and R10) between relapse and remission timepoints. How about the differentially expressed genes of one single memory B cell cluster between relapse and remission groups?*”

Response: Thank you for this important point. Differential gene expression analysis on R4 (MZ-like B cells) or R9 (SM cMBCs) subclusters yielded highly similar results to the differential gene expression analysis completed on all memory B cells in Fig. 6. Minimal differences in gene expression were observed between R10 B cells (atBCs) in Rel-RTX and Rem-RTX, which was expected as atBCs represent a more differentiated B cell population. However, since these differences do not arise at the pseudobulk level and memory B cells are scarce in post-RTX peripheral blood, we decided to provide a more robust analysis focusing on all MBCs instead.

6. Comment: “I am a little lost with the analyses in patients with a relapse after Rtx therapy. I understand that there is a difference between the flow cytometry analysis and the scRNA-seq (atBCs were unchanged in flow cytometry but MZ-like BCs, the predicted precursors of atBCs, were expanded in scRNA-seq analyses). How do the authors explain this? Is it because MZ-like cells cannot be analyzed using cytometry? Or is the scRNA-seq analysis is more sensitive? Or where these different patients? These circumstances are not entirely clear to me. Please better explain, also in the discussion.”

Response: The post-RTX scRNA-seq experiment outlined in Fig. 6 was done on four patients for whom we collected a post-RTX relapse (Rel-RTX) and post-RTX remission (Rem-RTX) sample (INS15, INS16, INS17, and INS18). By flow cytometry, we identified that a decreased frequency in transitional naïve B cells and an increased frequency in isotype-switched memory B cells was associated with Rel-RTX, a finding consistent with a previous report in pediatric INS (Colucci *et al.*, 2016 *Journal of the American Society of Nephrology*). This decreased frequency of transitional naïve B cells (cluster R1) and increase of isotype-switched memory B cells (cluster R9) was observed in three out of four children in post-RTX relapse (INS15, INS16, INS18) by scRNA-seq (**Fig. 6c**).

Strikingly, the most robust finding was the expansion of MZ-like B cells (cluster R4). There is no defined, specific marker panel to identify this population by flow cytometry. However, given their high *IGHM* expression and core MBC signature, they are likely contained within the isotype-unswitched (IgD⁺IgM⁺) cMBC population^{7,8}. Indeed, we detected an expansion of isotype-unswitched cMBC by flow cytometry within the total cohort (**Fig. 5h**) and in the scRNA-seq donors INS15-18 specifically (**Fig. R1**) in the Rel-RTX time point. **The relationship between MZ-like B cells detected by scRNA-seq and IgD⁺ IgM⁺ cMBC cells detected by flow cytometry was clarified in the main text as per the reviewer’s request (page 22).**

Figure R1: Frequency of IgM⁺ IgD⁺ cMBCs within the four donors used for the post-RTX scRNA-seq analysis determined flow cytometrically.

7. Comment: “*What could be the possible reasons for the relapse of INS patients after Rtx treatment, since the ASCs and atBCs are not significantly expanded in relapsed patients?*”

Response: This is an excellent point which we have put much thought in. We hypothesize that ASCs are the terminal effectors mediating pathology, likely through the production of anti-podocytopathic antibodies, a hypothesis that is in line with the recent finding of such antibodies in subsets of children with INS⁹. Moreover, we hypothesize that autoreactive B cells develop into ASCs through the extrafollicular route (e.g., during viral infection), largely through the rapid differentiation of atBCs¹⁰. The primary sites of these extrafollicular reactions are not within peripheral blood but rather within secondary lymphoid organs (spleens and lymph nodes)¹¹. Thus, while we did not observe an increase in atBC and ASCs in the peripheral blood during post-RTX relapse, these cells might be arising within secondary lymphoid organs representing a very early stage of immune re-activation and disease relapse. Consistently, this is further supported by the observation of MZ-like B cells within peripheral blood of Rel-RTX, a potential cellular source for extrafollicular autoreactive B cells¹². To test these hypotheses, identifying podocytopathic B cell clones from secondary lymphoid organs (e.g., tonsillectomy samples) would be desired and we are currently exploring how to obtain such samples.

8. Comment: “*In addition, did the naïve B cell clusters show some differences in gene expression profiles between the relapsed and remission samples?*”

Response: Pseudobulk differential gene expression analysis did not identify any differentially expressed genes between Rel-RTX and Rem-RTX in either the memory or naïve B cell clusters. Notably, this type of analysis highlights the difficulty of extracting relevant B cell information in post-RTX samples since 1) the peripheral blood is enriched with naïve B cells and 2) patients are at varying points along their B cell reconstitution. Hence, analyses were restricted to the memory compartment and were completed in a paired manner. Nevertheless, **we now provide standard differential gene expression analysis (without pseudobulk-level segregation) within the naïve B cell clusters, and we observed an enrichment of genes involved in B cell activation and type-I interferon signaling (Supplementary Fig. 9).**

It is important to note that our post-RTX analyses were limited by the fact that we were accessing samples in a non-interventional manner through an observational cohort. The Rel-RTX and Rem-RTX samples were, thus, obtained at various time points across the course of B cell reconstitution. To validate and strengthen our post-RTX analyses, we are aiming to collect blood samples from children at onset/relapse of INS without previous exposure to RTX, at remission after the first RTX dose during B cell recovery (at 4 months), and at relapse following the first

RTX dose (>6 months). Using scRNA-seq coupled with V(D)J-sequencing on FACS-isolated naïve and memory B cell subsets, we hope that this will provide a clearer picture at the pathogenic B cells in the post-RTX landscape.

9. Comment: “Some of the legends miss details on the statistical tests, sample numbers etc.”

Response: All legends were revised to clearly reflect the statistical tests used and sample numbers while maintaining the requirements indicated by the journal.

Reviewer #2 (expert in B cells in nephrotic syndrome):

1. Comment: “The selection of the patients is very rigorous and a detailed description of the characteristics of each patient is reported in supplementary Data 1; however, this description cannot be easily extrapolated and details on the state of the disease as well as on the current medication at time of sampling is not so clear by reading the text. I would suggest to add a Table resuming patient characteristic sub-grouped as described in figure 5a (INS, Rem-GC, Rel-RTX, Rem-RTX) by reporting the most relevant info (mean-SD age, # of patients at onset or in relapse for patients in active phase, # of patients on current or previous medication, mean time from last medication when administered (mainly for GC and RTX), mean-SD uPCR, co-morbidity, RNAseq or flow cytometry analysis) for each subgroup.”

Response: A Table containing the requested information is now provided (Supplementary Table 1) on page 50. The time since last prednisone administration is not included as this information is difficult to obtain with accuracy for children who were prescribed their medication by telephone and children with poor adherence to medication. However, the time since the last RTX dose is now provided as children were brought into the hospital to receive the infusion. **This information was also added to Supplementary Data 1 for every donor.**

2. Comment: “What are the inclusion criteria for healthy donors? Where are they enrolled? Are they patients followed at the Institution for other non-immunological conditions? Please specify.”

Response: Healthy children undergoing minor day surgery or healthy volunteers from the community (e.g., the children of the Research Institute personnel) were enrolled to serve as healthy controls. **This is now included in the main text of the paper on pages 24-25.**

3. Comment: “For patients treated with RTX, relapsing patients have a re-mergence of isotype-switched memory B cells more than atypical memory B cells or ASCs. In addition to the hypothesis reported in the current manuscript, this association could be due to the persistence of isotype-switched and clonally expanded memory B cell clones, as previously reported in other autoimmune diseases (Bashford-Rogers RJM, et al. Nature. 2019. doi:10.1038/s41586-019-1595-3). This could also be due to the persistence of isotype-switched memory B cell clones in secondary lymphoid organs, which are not effectively depleted by rituximab treatment (Wallin et al. Blood. 2014. doi: 10.1182/blood-2014-07-585976). Both papers should

be commented and reported in the discussion. In this regard, the characterization of the BCR repertoire as performed by Bashford-Rogers et al. in longitudinal samples could be very helpful to verify also in the setting of INS whether the post-RTX re-emerging B cell subsets are persistent or newly emerging clones.”

Response: We thank the reviewer for this insightful comment which we agree with. While we did not identify an increase in atBCs in post-RTX relapse samples, they may contribute to a pool of autoreactive isotype-switched B cell clones persisting in secondary lymphoid organs. Indeed, the INS-associated atBCs we identified were largely isotype-switched cells (IgD⁻ IgM⁻) (**Fig. 3b, c, 4b**). As extrafollicular reactions do not primarily originate within peripheral blood, we are likely observing a small portion of this response which predominantly takes place within secondary lymphoid tissues¹¹. Hence, we were not surprised with the lack of observed differences in proportions/numbers of atBCs and ASCs in post-RTX peripheral blood. The expansion of MZ-like B cells in peripheral blood (cluster R4 in **Fig. 6a**, and IgM⁺ IgD⁺ cMBCs in **Fig. 5h**), an extrafollicular B cell population enriched with autoreactive BCRs¹³ in post-RTX relapse supports this point. To test this hypothesis, we would like to evaluate the BCR repertoire in paired tonsillectomy-PBMC samples from children with active INS and pre/post-RTX treatment, while recognizing the challenge to access such samples. **The possibility of clonally expanded persistent memory B cell clones in participating in the pathogenesis of post-RTX relapse has been discussed in the main text on page 22.**

4. Comment: *“The amount of total B cells, as determined by flow cytometry, should be reported by comparing INS patients and HC, to verify the results obtained by RNAseq.”*

Response: While we did not observe an increase in the proportion of total B cells in children with active INS in the cohort, we did observe a specific increase in isotype-switched cMBCs, CD21^{low} B cells, and ASCs (**Fig. 3c**) corresponding with increases in memory B cells observed in scRNA-seq (**Supplementary Fig. 2b**). **The proportion of total B cells is now provided in Supplementary Fig. 6a.**

5. Comment: *“In supplemental figure 4a is reported the intensity of CD19 expression on the surface of CD21^{low} B cells. Why the expression of CD19 is significantly lower in INS patients compared to HC? Please describe and discuss it.”*

Response: This is a very interesting observation. High CD19 expression is a key feature of atBCs which was first described by scRNA-seq^{5,6}. Our own characterization of atBCs by scRNA-seq confirms the highest expression of *CD19* within this population (**Fig. 2d**). The atBC population defined by scRNA-seq best corresponds with T-bet⁺ CD11c⁺ CD21^{low} B cells defined by flow cytometry. These cells were the specific CD21^{low} B cell population expanded in INS (**Fig. 4b**). In this T-bet⁺ CD11c⁺ CD21^{low} population, high CD19 expression was evident in both HC and INS individuals (**Supplementary Fig. 6b**). However, as Reviewer 2 points out, when looking at total CD21^{low} B cells, the expression of CD19 in this population is diminished in INS in comparison to

HC (**Supplementary Fig. 6b**). We predict that this is due to the loss of CD21 expression in B cells that have not yet gained T-bet and CD11c expression in INS, a phenomenon that correlates with chronic inflammation and autoimmunity. **Fig. 4 and Supplementary Figure 6b (previously Supplementary Figure 4a) have been updated for clarification.**

Reviewer #3 (expert in B cells in autoimmune disease):

1. Comment: “Fig 1 shows an INS signature. It probably should be noted how much of this is a consequence of a different frequency of B cells (shown in Fig 2) rather than DEGs within subsets.”

Response: This comment was also raised by Reviewer #4. We have revised Figure 2 to directly address this question. Undoubtedly, a portion of the nephrotic signature is likely driven by the expansion of memory cells and the contraction of naïve cells. Nevertheless, pseudobulk differential gene expression analysis performed on each B cell subcluster demonstrates that each naïve and memory B cell cluster in INS individuals, apart from atBCs and ASCs, preferentially express genes involved in B cell activation and actin cytoskeleton rearrangement, the predominant pathways upregulated in the nephrotic B cell signature (**Supplementary Fig. 5b**). Moreover, Naïve-1, actMBCs, MBC-2, and MZ-like B cells in INS preferentially expressed genes associated with the formation of the extrafollicular responses (e.g., *TNFRSF13B*, *GPR183*, *POU2F2*, *POU2AF1*) (**Fig. 2e**). Thus, we conclude that the nephrotic B cell signature is driven by the frequency of B cell subsets, and also by transcriptional differences within each subset.

2. Comment: “The study of those who relapse after rituximab is of great interest. The authors discuss the difference in frequency of MZ B cells. They might want to focus on the difference in frequency of transitional and naïve B cells as this probably reflects BAFF levels and belimumab might be of therapeutic use.”

Response: Indeed, transitional naïve B cells are diminished during active INS in comparison to HC (**Fig. R2**). In the post-RTX timepoint, relapses were also associated with a decrease in transitional naïve B cells (**Fig. 5g**). However, post-RTX relapses were still enriched with transitional naïve B cells when compared to HC or active INS suggesting that post-RTX peripheral blood is highly enriched with antigen-inexperienced populations making the investigation of pathogenic/antigen-specific B cell subsets very difficult. Nevertheless, we aimed to investigate transitional naïve B cells in RTX-unexposed individuals with active INS by conducting pseudobulk differential gene expression analysis within the transitional naïve B cell population (**Fig. 2e**). While we did not observe upregulation of BAFF receptors *TNFRSF13B* and *TNFRSF13C*, transitional naïve B cells in INS preferentially expressed genes associated with B cell activation and antigen presentation (**Supplementary Fig. 5b**) denoting baseline differences in the transitional naïve B cell population between active INS and HC.

Figure R2: Frequencies of transitional naïve B cells in HC and INS individuals determined flow cytometrically.

Reviewer #4 (expert in single-cell RNA sequencing):

Major points

1. Comment: *“Absence of single cell quality controls: There is no information given to prove the donor/sample quality are comparable, such as the number of genes/UMIs detected and percentage of mitochondrial genes. This information is important for any downstream analysis and comparison.”*

Response: We thank Reviewer 4 for his/her thorough review and comments. The sample quality was comparable across all samples, although the frequency of mitochondrial genes was elevated in sample INS2 leading to more cells being removed in QC steps. Interestingly, all INS samples had greater frequencies of doublets identified when removing cells expressing at least two distinct lineage-defining genes (*CD79A*, *CD3G*, *CD14*, and *LILRA4*) and by using DoubletFinder. As these samples were processed independently for biobanking, and since all samples were processed at the same time for scRNA-seq analysis, we think that this is a true biological effect. **Supplementary Figure 1 was updated with the requested QC metrics.**

Comment: *“For example, in Fig. 1B the author compared the major cell lineage proportions between INS and HC and concluded the major proportion difference is B cells, while it showed one single sample is dominating the comparison for B cells.”*

Response: Reviewer 4 is correct that the INS2 sample is driving the increase in proportions of total B cells. When we looked at total B cell proportions by flow cytometry, we did not observe

a difference in total B cells between HC and INS (**Supplementary Fig. 6a**). **We now provide the proportions of each individual PBMC cluster in Supplementary Fig. 2b.** As is evident, in all children with INS, the proportions of memory B cells (C11 and C12) and ASCs (C13) are much greater than in HC. The strength of this finding remains true even if INS2 is removed: all the analyses shown in this manuscript hold true even with the removal of INS2. Thus, the predominant immune defect in INS is indeed the expansion of memory B cells.

2. Comment: *“Missing information on unbiased cluster markers: For both of the two clustering annotation analyses (PBMC and B cells), the authors didn’t provide the top marker genes lists for all the clusters. For all PBMC clusters, the authors only showed selected canonical lineage markers in Fig. S1B but did not optimize the order of the markers. For the subclusters of B cells, the marker genes were missing. The authors showed the heatmap for memory cell clusters (Fig. 2D), but not for other B cell clusters (B 0,4,7,8,9). Please include the complete top markers of all clusters in Fig. S3.”*

Response: We appreciate this comment and have adjusted the manuscript accordingly. **We now provide a heatmap in Supplementary Fig. 2c showing the top 10 markers expressed in each PBMC cluster.** The expression of select canonical lineage markers is shown in feature plots in Supplementary Fig. 2d for easy understanding of the UMAP. **Similarly, a heatmap in Supplementary Fig. 4a shows the top 10 markers expressed in each B cell subcluster.** We kept the heatmap comparing only memory B cell subclusters in Fig. 2d to highlight specific genes of interest (e.g., *TNFRSF13B*, *GPR183*, *POU2F2*, *CD19*, *FCRL5*, etc.).

3. Comment: *“For the pseudobulk differential gene expression analysis, the authors should also set a p-value threshold to determine the significantly changed genes instead of only using log2FC and percentage of expression.”*

Response: An adjusted p-value threshold of less than 0.05 was indeed used in this analysis. **This was mistakenly omitted from the text and is now included.**

4. Comment: *“Fig. S1C showed that only 381 differentially expressed genes were unique for B cells, the remaining 577 genes were shared between B cells and other cell types. For the pathway analysis, the authors should explore which pathways are unique for B cells and which ones are shared between B cells and other immune cells. This is the advantage of this nice single cell dataset and should not be ignored.”*

Response: We appreciate the reviewer’s comment. We conducted pathway analysis on the 250 genes that were specifically upregulated in B cells (**Fig. R3**). The observations made here were very similar to those observed using all B cell genes although the analysis was enriched with terms involved in B cell activation and humoral immunity, and fewer terms associated with actin cytoskeleton dynamics. Some of the key genes upregulated specifically in B cells included genes associated with B cell activation (e.g., *SPI1*, *POU2AF1*, *BANK1*, and *BLNK*), atBC-associated genes (e.g., *FCRL1*, *FCRL2*, *ENC1*, and *CD19*), MZ-like B cell-associated genes (*MZB1*,

TNFRSF13B, *FCRLA*, and *FCGR2B*), and immunoglobulin genes. We used the analysis on all B cell genes in the manuscript to give a more holistic view of the nephrotic B cell signature, especially since we anticipated that certain pathways would be conserved across immune cell subsets. For genes that were upregulated in all PBMC subsets, pathway analysis showed minimal enrichment of terms associated with intracellular protein transport.

Figure R3: Gene Ontology hits from pathway analysis conducted on the B cell-specific genes that are upregulated in INS.

5. Comment: “The analysis in Fig. 2E is not convincing. The nephrotic signatures were calculated on pseudobulk B cells. And the bigger clusters within the B cells would contribute more to the global B cell nephrotic signatures. It seems that B3 and B5 have more cells than B6. So it is not entirely fair to say that the nephrotic signature was most pronounced in MZ-like B cells (B3) and SM B cells (B5). One alternative approach is to perform differential expression analysis between healthy and disease on each subcluster of B cells and then the enrichment of pathways for each subcluster.”

Response: This comment by Reviewer 4 is justified and well-taken. We ran pseudobulk differential gene expression analysis for each B cell subcluster to identify the genes perturbed in INS compared to HC (**Fig. 2e, Supplementary Fig. 5a**). In all cases, INS B cells expressed genes associated with actin cytoskeleton rearrangement (**Supplementary Fig. 5b**). The Naïve-1, actMBC, and MZ-like B cell subclusters also showed an enrichment of genes associated with B cell activation and antigen presentation (**Supplementary Fig. 5b**). The pathways and genes upregulated in these clusters were salient features of the nephrotic B cell signature demonstrating that this signature did not arise from a fluctuation in the frequencies of certain B cell subsets, but rather by key changes at the transcriptional level. Moreover, the Naïve-1, actMBC, and MZ-like B cell subclusters also expressed genes associated with extrafollicular responses including *TNFRSF13B* (the most highly differentially expressed gene within the nephrotic B cell signature), *GPR183*, *POU2F2* and *POU2AF1*. **To satisfy the concern of Reviewer 4, Figure 2 has been adjusted accordingly.**

6. Comment: “The authors compared the features of CD21^{low} B cells identified in flow cytometry data to that of atBCs recovered from scRNA-seq data. It’s not clear what the conclusion is. A better description and visualization should be provided.”

Response: We apologize for the lack of clarity on this key point of our study. We now provide a more streamlined analysis of the atBC population by flow cytometry. In Figure 3, we used FlowSOM using ten B cell markers to show that INS was associated with cycling (Ki-67⁺) isotype-switched (IgM⁻ IgD⁻) CD21^{low} B cells. These CD21^{low} B cells are associated with B cell dysfunction in several autoimmune, chronic inflammatory, and immunodeficient conditions¹⁴⁻¹⁹. These cells also lacked CXCR5 expression indicating that these CD21^{low} B cells are an extrafollicular B cell population. Since atBCs consist of an extrafollicular population defined by multiple markers, including T-bet and CD11c co-expression²⁰⁻²², and are also CD21^{low} B cells, we hypothesized that the expansion of CD21^{low} B cells observed in Fig. 3 was closely associated with an expansion of atBCs. As such, we show that atBCs (T-bet⁺ CD11c⁺) are almost exclusively found in the CD21^{low} B cell compartment with the frequency of CD21^{low} B cells directly correlating with the frequency of T-bet⁺ CD11c⁺ B cells (**Fig. 4a**). These atBCs were indeed expanded in INS (**Fig. 4b**) and this is more striking when we focus our analysis onto isotype-switched atBCs. We then show that atBCs in INS have greater expression of T-bet, which is associated with a greater capacity to convert into ASCs²³, and lower expression of FcRL5, which is associated with dysregulated BCR signaling²⁴. **Figure 4 and the text (pages 13-14) were revised to clarify this important point.**

7. Comment: “The clustering resolutions used for different scRNA-seq analysis were all set to be 0.5. However, the separation was not optimized especially for Fig. 6. The authors should consider trying different resolutions and optimize the clustering and the annotation.”

Response: All resolutions between 0.4-2.0 (at increments of 0.1) were performed for the clustering of total PBMC (Fig. 1), B cell subclusters (Fig. 2), and the post-RTX B cells (Fig. 6). In Fig. 6, given the naïve-enriched post-RTX B cell landscape, greater levels of resolution generated too many naïve B cell sub-populations while memory B cell populations remained as four distinct clusters (MZ-like B cells, atBC, SM, ASC). At lower resolutions, the memory populations merged. As such, we maintained the 0.5 clustering resolution despite the greater number of cells within the sample. The figure below shows clustering at 0.2, 0.5, 1.0, 1.5, and 2.0 (**Fig. R4**).

Figure R4: UMAP clustering of post-RTX B cells at different resolutions.

8. Comment: “It would be great if the authors could validate some of the scRNA-seq results using e.g. qPCR.”

Response: We thank Reviewer 4 for this comment. While this is something that we plan for future studies, we were unable to validate our findings by qPCR as these PBMC samples are limited in number and abundance. Thus, we focused our attention on flow cytometric B cell characterization in greater numbers of children with INS and HCs to show and validate the extrafollicular differentiation of B cells involving atBCs and short-lived ASCs. Future work will be aimed at investigating the role and relevance of *TNFRSF13B* overexpression in INS B cells, and the possibility of targeting BAFF and APRIL signaling for therapeutic purposes.

Minor points

1. Comment: “The two UMAPs in Fig. S1A are not sufficient to demonstrate that the 18 distinct immune cell populations were uniformly present in all donors. This can only validate that all the populations are present at both disease and healthy conditions. Please provide a barplot of the cluster distribution across all the donors.”

Response: Box plots showing the cluster distribution across all donors is now provided in Supplementary Fig. 2b.

2. Comment: “Cluster C1 in Fig. 1A and Fig. S1B showed a clear expression of Treg signatures (FOXP3, CTLA4). The current annotation of T-mem1 is less accurate.”

Response: Reviewer 4 is correct to note that the C1 cluster contains regulatory T cells. However, these cells represent less than 5% of cells within this cluster, especially in children with INS. A decrease in Treg cells is a well-established feature of active INS. While we had hoped to perform in-depth analysis of regulatory T cells using this scRNA-seq dataset, we do not have enough cells here. Future investigation will target the transcriptional regulatory T cell defect in pediatric INS. However, since the C1 cluster is composed largely of memory T cells (including regulatory T cells), we named this cluster Tmem-1.

3. Comment: “Fig. S1C only shows the upregulated intersection of genes, but the figure legend doesn’t mention that.”

Response: The figure was generated using the UpSetR package using the map() function in R. The intersection of genes shows all the genes that are differentially expressed (either upregulated or downregulated) as per the $|\log_2FC| > 0.65$ and $p < 0.05$ thresholds.

4. Comment: “It would be great if the authors could reproduce the transcription factor enrichment analysis using SCENIC and provide the results in a supplementary figure.”

Response: Indeed, transcription factor analysis using SCENIC would be very interesting and would confirm our current results. We will perform this new analysis soon.

5. Comment: “In Fig. 2G the three trajectory branches are not clear.”

Response: It should be noted that we confirmed this trajectory using Monocle2 as well as Slingshot at multiple resolutions. We have slightly modified the trajectory analysis by generating a new UMAP based on all B cell subclusters except for ASCs (as the previous trajectory was built on the UMAP including ASCs). **The lines of the trajectory were placed on this new UMAP and a separate UMAP showing pseudotime values is now provided.**

6. Comment: “The authors used “CellChat” for APRIL signaling network analysis in Fig. 2H but the corresponding methods section is missing.”

Response: The methods section has been updated to include the use of the CellChat database for cell network analysis (page 28).

7. Comment: In Fig. 3A the FlowSOM clustering result is difficult to read (especially the text coloring is hard to take in).

Response: This comment by Reviewer 4 is well-taken. We added a gray background to the FlowSOM to accentuate the clarity of the text. Unfortunately, we are unable to change the colour palette of the FlowSOM maps as this is a very new feature of the flow cytometry analysis software FlowJo and color changes are not possible.

We would like to thank all Reviewers and the editorial team for their suggestions in improving the clarity of our manuscript, through both experimental additions, as well as changes to our explanations in the body of the text. We feel we have fully addressed all the comments and concerns raised by the Reviewers. If you have additional comments, please feel free to contact us at any time. Thanks to the reviewers, our revised manuscript is now stronger in nature. We believe that our study now merits publication in *Nature Communications*.

References

- 1 Yang, C. *et al.* Heterogeneity of human bone marrow and blood natural killer cells defined by single-cell transcriptome. *Nat Commun* **10**, 3931 (2019). <https://doi.org:10.1038/s41467-019-11947-7>
- 2 Ye, Q. *et al.* The immune cell landscape of peripheral blood mononuclear cells from PNS patients. *Scientific Reports* **11**, 13083 (2021). <https://doi.org:10.1038/s41598-021-92573-6>
- 3 Fehniger, T. A. *et al.* CD56bright natural killer cells are present in human lymph nodes and are activated by T cell-derived IL-2: a potential new link between adaptive and innate immunity. *Blood* **101**, 3052-3057 (2003). <https://doi.org:10.1182/blood-2002-09-2876>
- 4 Portugal, S. *et al.* Malaria-associated atypical memory B cells exhibit markedly reduced B cell receptor signaling and effector function. *Elife* **4** (2015). <https://doi.org:10.7554/eLife.07218>
- 5 Sutton, H. J. *et al.* Atypical B cells are part of an alternative lineage of B cells that participates in responses to vaccination and infection in humans. *Cell Rep* **34**, 108684 (2021). <https://doi.org:10.1016/j.celrep.2020.108684>
- 6 Holla, P. *et al.* Shared transcriptional profiles of atypical B cells suggest common drivers of expansion and function in malaria, HIV, and autoimmunity. *Sci Adv* **7** (2021). <https://doi.org:10.1126/sciadv.abg8384>
- 7 Bautista, D. *et al.* Differential Expression of IgM and IgD Discriminates Two Subpopulations of Human Circulating IgM(+)IgD(+)CD27(+) B Cells That Differ Phenotypically, Functionally, and Genetically. *Front Immunol* **11**, 736 (2020). <https://doi.org:10.3389/fimmu.2020.00736>
- 8 Siu, J. H. Y. *et al.* Two subsets of human marginal zone B cells resolved by global analysis of lymphoid tissues and blood. *Science Immunology* **7**, eabm9060 (2022). <https://doi.org:doi:10.1126/sciimmunol.abm9060>
- 9 Watts, A. J. B. *et al.* Discovery of Autoantibodies Targeting Nephritin in Minimal Change Disease Supports a Novel Autoimmune Etiology. *Journal of the American Society of Nephrology* **33**, 238 (2022). <https://doi.org:10.1681/ASN.2021060794>

- 10 Jenks, S. A., Cashman, K. S., Woodruff, M. C., Lee, F. E. & Sanz, I. Extrafollicular responses in humans and SLE. *Immunol Rev* **288**, 136-148 (2019). <https://doi.org:10.1111/imr.12741>
- 11 Lam, J. H., Smith, F. L. & Baumgarth, N. B Cell Activation and Response Regulation During Viral Infections. *Viral Immunol* **33**, 294-306 (2020). <https://doi.org:10.1089/vim.2019.0207>
- 12 Weller, S. *et al.* Human blood IgM "memory" B cells are circulating splenic marginal zone B cells harboring a prediversified immunoglobulin repertoire. *Blood* **104**, 3647-3654 (2004). <https://doi.org:10.1182/blood-2004-01-0346>
- 13 Palm, A.-K. E. & Kleinau, S. Marginal zone B cells: From housekeeping function to autoimmunity? *Journal of Autoimmunity* **119**, 102627 (2021). <https://doi.org:https://doi.org/10.1016/j.jaut.2021.102627>
- 14 Claes, N. *et al.* Age-Associated B Cells with Proinflammatory Characteristics Are Expanded in a Proportion of Multiple Sclerosis Patients. *J Immunol* **197**, 4576-4583 (2016). <https://doi.org:10.4049/jimmunol.1502448>
- 15 Warnatz, K. *et al.* Expansion of CD19(hi)CD21(lo/neg) B cells in common variable immunodeficiency (CVID) patients with autoimmune cytopenia. *Immunobiology* **206**, 502-513 (2002). <https://doi.org:10.1078/0171-2985-00198>
- 16 Wehr, C. *et al.* A new CD21low B cell population in the peripheral blood of patients with SLE. *Clin Immunol* **113**, 161-171 (2004). <https://doi.org:10.1016/j.clim.2004.05.010>
- 17 Wildner, N. H. *et al.* B cell analysis in SARS-CoV-2 versus malaria: Increased frequencies of plasmablasts and atypical memory B cells in COVID-19. *J Leukoc Biol* **109**, 77-90 (2021). <https://doi.org:10.1002/jlb.5cova0620-370rr>
- 18 Isnardi, I. *et al.* Complement receptor 2/CD21– human naive B cells contain mostly autoreactive unresponsive clones. *Blood* **115**, 5026-5036 (2010). <https://doi.org:10.1182/blood-2009-09-243071>
- 19 Malle, L. *et al.* Autoimmunity in Down's syndrome via cytokines, CD4 T cells and CD11c+ B cells. *Nature* **615**, 305-314 (2023). <https://doi.org:10.1038/s41586-023-05736-y>
- 20 Wang, S. *et al.* IL-21 drives expansion and plasma cell differentiation of autoreactive CD11c(hi)T-bet(+) B cells in SLE. *Nat Commun* **9**, 1758 (2018). <https://doi.org:10.1038/s41467-018-03750-7>
- 21 Csomos, K. *et al.* Partial RAG deficiency in humans induces dysregulated peripheral lymphocyte development and humoral tolerance defect with accumulation of T-bet(+) B cells. *Nat Immunol* **23**, 1256-1272 (2022). <https://doi.org:10.1038/s41590-022-01271-6>
- 22 Keller, B. *et al.* The expansion of human T-bet(high)CD21(low) B cells is T cell dependent. *Sci Immunol* **6**, eabh0891 (2021). <https://doi.org:10.1126/sciimmunol.abh0891>
- 23 Stone, S. L. *et al.* T-bet Transcription Factor Promotes Antibody-Secreting Cell Differentiation by Limiting the Inflammatory Effects of IFN- γ on B Cells. *Immunity* **50**, 1172-1187.e1177 (2019). <https://doi.org:10.1016/j.immuni.2019.04.004>
- 24 Haga, C. L., Ehrhardt, G. R. A., Boohaker, R. J., Davis, R. S. & Cooper, M. D. Fc receptor-like 5 inhibits B cell activation via SHP-1 tyrosine phosphatase recruitment. *Proceedings of the National Academy of Sciences* **104**, 9770-9775 (2007). <https://doi.org:doi:10.1073/pnas.0703354104>

REVIEWERS' COMMENTS

Reviewer #1 (expert in nephrology):

The authors have extensively revised the manuscript and addressed all of my comments appropriately. Thank you very much.

This is an excellent and elegant study, congratulations.

I have one last minor comment:

Please increase the resolution of the panels in Supplementary Figure 5b to allow a clear identification and evaluation of the identified pathways.

Reviewer #2 (expert in B cells in nephrotic syndrome):

The manuscript has been improved based on reviewers' comments. I have no further question or comment.

Reviewer #3 (expert in B cells in autoimmune disease):

I am satisfied with the responses to the review. This is a very important and well executed study.

Reviewer #4 (expert in single-cell RNA sequencing):

In general the authors did a reasonable job revising the manuscript. I think the main findings hold, although I have some minor remarks regarding the computational analysis of the data.

The proper way to analyze the INS signature would be to fit a linear model including the proportion shifts as covariates, which could partially disentangle the effects of varying amounts of cell subsets and general DEGs. Moreover and more importantly, this would result in more readily interpretable p-values.

Response to reviewer comments

We are happy to hear your willingness to accept our manuscript entitled “*The extrafollicular B cell response is a hallmark of childhood idiopathic nephrotic syndrome*” (Ms. No. NCOMMS-23-23131) for publication in *Nature Communications*. We thank the reviewers for their excellent suggestions for revisions which have collectively improved the quality of our submission.

Here, we submit a revised version of our manuscript that incorporates all the edits and recommendations that reviewers have made. In particular:

- We included all statistical information in figure legends and in the “*Statistical analysis*” section of our manuscript, and absolute *P*-values were provided in all the figures in place of the asterisks used earlier.
- A more complete statement regarding informed consent is provided in the manuscript. As you correctly pointed out, the children themselves were not consented but rather parents/legal guardians. In addition to this, children between the ages of 7 and 18 years were asked to sign an assent form.
- The font sizes in figures have been increased wherever possible and high quality vectorized images are provided. Please note that the heatmap in Figure 6b is too large to be provided as a vector (369MB) and is therefore provided as a high resolution .png file. If the PDF submitted is not sufficient quality, we are happy to provide the PPT file for the entire figure.
- Supplementary Figure 6 was split into two figures to include flow cytometry gating strategies. Supplementary Figure 6 shows the gating of FlowSOM populations (Fig. 3) using traditional B cell gates. Supplementary Figure 7 shows the gating using our atypical B cell panel (Fig. 4).
- Our data deposited in GEO is now publicly available.
- Please note that we have not used any custom code in this manuscript. All code was based on code generated by package developers. All the scripts used to generate the Seurat objects, run the pseudobulk differential gene expression analysis (with Muscat), trajectory analysis (with Monocle3), and cell network analysis (with CellChatDB) are ready to be submitted.
- The nature reporting summary was updated to include a statement on code availability, replication, and blinding. The “*Data exclusions*” of the “*Life sciences study design*” was updated to mention that doublets and low-quality cells were removed from the Seurat objects, as stated in the methods of the manuscript. The flow cytometry section was also updated to reflect that the figures now include axis scales for flow plots and gating strategies.

Once again, we thank the Reviewers for a very positive review experience, their thorough and thoughtful comments, and numerous constructive suggestions for the improvement of our submission. The individual comments made by the Reviewers are addressed point-by-point below (page 2). At the end of this document, we also provide the summary requested in the Author Checklist (page 3).

Reviewer #1 (expert in nephrology):

1. Comment: *“The authors have extensively revised the manuscript and addressed all of my comments appropriately. Thank you very much. This is an excellent and elegant study, congratulations.”*

Response: Thank you for the kind words. We appreciate all the insightful comments that you provided us in the review process. We are delighted to have the opportunity to share this story with the greater nephrology community.

2. Comment: *“Please increase the resolution of the panels in Supplementary Figure 5b to allow a clear identification and evaluation of the identified pathways.”*

Response: We now provide a high-resolution version of Supplementary Figure 5b. We also increased font sizes here (and in all other figures).

Reviewer #2 (expert in B cells in nephrotic syndrome):

1. Comment: *“The manuscript has been improved based on reviewers’ comments. I have no further question or comment.”*

Response: Thank you for your deep and critical analysis of our manuscript.

Reviewer #3 (expert in B cells in autoimmune disease):

1. Comment: *“I am satisfied with the responses to the review. This is a very important and well executed study.”*

Response: We appreciate the kind words. Thank you for your thorough review of our manuscript.

Reviewer #4 (expert in single-cell RNA-sequencing):

1. Comment: *“In general the authors did a reasonable job revising the manuscript. I think the main findings hold, although I have some minor remarks regarding the computation analysis of the data. The proper way to analyze the INS signature would be to fit a linear model including the proportion shifts as covariates, which could partially disentangle the effects of varying amounts of cell subsets and general DEGs. Moreover, and more importantly, this would result in more readily interpretable p-values.”*

Response: We appreciate the thorough and critical review of our manuscript. Undoubtedly, the recommendations made by the reviewer led to clearer interpretation of our scRNA-seq data. In future studies, we will strongly consider the suggestion made by the reviewer to determine the importance of DEGs independently of shifts in the proportions of cell subsets.

Summary (246 words)

Idiopathic nephrotic syndrome (INS) is the most common chronic kidney disease affecting children. Children afflicted with this condition endure recurrent episodes of nephrotic-range proteinuria, where essential blood proteins are lost in the urine, resulting in debilitating edema, often coinciding with acute respiratory viral infections. Although INS is controlled well with broad immunosuppression, as much as 80% of children experience relapses, necessitating additional rounds of treatment and causing significant treatment-related side effects. Consequently, there is a pressing need of safer and more targeted treatments. The recent success of B cell-depleting therapeutics like rituximab at maintaining a proteinuria-free status and the identification of kidney-targeting autoantibodies in subsets of children with INS has suggested the involvement of antibody-producing B cells in the pathogenesis of INS. Nevertheless, the precise nature of the immune response giving rise to these pathogenic B cells remained elusive. Using single-cell RNA-sequencing of peripheral blood, Al-Aubodah and colleagues demonstrate that the predominant peripheral immune cell defect in INS is an overtly activated B cell pool. Extrafollicular memory B cells, which rapidly emerge during viral infections, were notably expanded in INS. The authors confirm the expansion of extrafollicular memory B cells as T-bet⁺CD11c⁺ B cells and rituximab-sensitive T-bet⁺ plasmablasts, two populations that have recently been implicated with autoimmunity, in a greater number of children with INS. These findings substantiate a B cell-dependent origin for INS and propose a key link between autoantibody development and the co-occurrence of relapses with viral infections through the extrafollicular B cell response.